# High-Throughput Strategies for the Design, Discovery, and Analysis of Bismuth-Based Photocatalysts

**DOI:** 10.3390/ijms24010663

**Published:** 2022-12-30

**Authors:** Surya V. Prabhakar Vattikuti, Jie Zeng, Rajavaram Ramaraghavulu, Jaesool Shim, Alain Mauger, Christian M. Julien

**Affiliations:** 1School of Mechanical Engineering, Yeungnam University, Gyeongsan 38541, Republic of Korea; 2Department of Physics, School of Applied Sciences, REVA University, Bangalore 560064, India; 3Institut de Minéralogie, de Physique des Matériaux et de Cosmochimie (IMPMC), Sorbonne Université, CNRS-UMR 7590, 4 Place Jussieu, 75252 Paris, France

**Keywords:** bismuth materials, photocatalysts, solar energy, surface engineering, visible light, nanostructure

## Abstract

Bismuth-based nanostructures (BBNs) have attracted extensive research attention due to their tremendous development in the fields of photocatalysis and electro-catalysis. BBNs are considered potential photocatalysts because of their easily tuned electronic properties by changing their chemical composition, surface morphology, crystal structure, and band energies. However, their photocatalytic performance is not satisfactory yet, which limits their use in practical applications. To date, the charge carrier behavior of surface-engineered bismuth-based nanostructured photocatalysts has been under study to harness abundant solar energy for pollutant degradation and water splitting. Therefore, in this review, photocatalytic concepts and surface engineering for improving charge transport and the separation of available photocatalysts are first introduced. Afterward, the different strategies mainly implemented for the improvement of the photocatalytic activity are considered, including different synthetic approaches, the engineering of nanostructures, the influence of phase structure, and the active species produced from heterojunctions. Photocatalytic enhancement via the surface plasmon resonance effect is also examined and the photocatalytic performance of the bismuth-based photocatalytic mechanism is elucidated and discussed in detail, considering the different semiconductor junctions. Based on recent reports, current challenges and future directions for designing and developing bismuth-based nanostructured photocatalysts for enhanced photoactivity and stability are summarized.

## 1. Introduction

Making the most of renewable resources is a top priority. In this regard, the use of solar energy as a perennial energy source to drive chemical transformation is at the forefront of this movement. In recent years, energy demand and crisis have experienced tremendous growth due to the gradual growth of the global population and industrialization [1]. Primary energy sources come from fossil fuels, including coal, oil, and natural gas. However, one of the main problems associated with the consumption of these non-renewable energy sources is that they will run out soon. Another downside associated with this major energy consumption is the emission of carbon dioxide, which contributes significantly to global warming. Therefore, it is mandatory for humanity to switch to renewable energy sources and reduce greenhouse gas emissions. In this regard, hydrogen (H_2_) is a completely clean energy source when it is obtained from water-splitting processes and has been reported as an alternative energy source [2,3]. The coming decades will experience sustainable growth in its production and consumption. The photocatalytic process utilizing abundant solar energy as a light source is considered the key to hydrogen production and carbon dioxide conversion [4]. In addition, photocatalysis can also be used to deal with some other environmental problems, including the degradation of organic pollutants, such as rhodamine B (RhB), indigo carmine (IC), methylene blue (MB), methyl orange (MO), brilliant green (BG), bisphenol A (BPA), tetracycline (TC), chlortetracycline (CTC), acetaminophen (APAP), 4-tert-butylphenol (BTBP), ciprofloxacin (CIP), 2,4-dichlorophenol (2,4-DCP), etc., through some complex reactions under mild conditions, such as photocatalytic organic synthesis [5]. Photocatalysis has emerged as a benchmark tool that combines light and catalysts to perform chemical transformations that are elusive using standard synthetic procedures.

In the past decade, rapid development in the synthesis of photocatalyst nanomaterials has taken place [6]. In addition to this, a variety of photocatalysts have been used, which can convert sunlight into chemical energy and transfer this energy to reactive molecules for efficient photo-fuel conversion. Many catalysts have been studied for degrading pollutants or water splitting. Wide-gap semiconductors such as TiO_2_ can absorb only <5% of solar light. On the other hand, small-gap semiconductors can absorb visible light, but their photocatalytic performance is also poor because of the rapid electron-hole recombination rate. Bismuth compounds have emerged as a family of promising photocatalysts. According to their composition, their band gap can be adjusted to the desired value for visible light absorption. In addition, their internal electrical field induces the separation of the photogenerated charge carriers so that the recombination rate is reduced. Among the different photocatalyst groups, photocatalysts based on nanostructures, nanocomposites, or heterostructures have shown superior photodegradation efficiency compared to their bulk counterparts [7,8]. However, the relevant photocatalytic processes involved in nanostructured photocatalysts are more complex and far from fully understood. For example, identifying the real catalytically active species remains unsolved in most cases and is needed to obtain reproducible and efficient nanostructured or heterostructured photocatalyst materials for practical applications. In fact, the importance of discovering the truly catalytically active species involved in photocatalytic systems allows for a better and more general understanding of the photocatalytic process, which can help improve its efficiency.

In this review article, more attention is paid to the fundamental concepts, mechanisms of the synthetic process, and structural features dependent on the photocatalytic activity of bismuth-based photocatalysts. Their intrinsic photocatalytic properties, namely the charge transfer and separation, excitation formation, and catalytic activity by the formation of nanostructures are described. Their recent trends, including photodegradation, H_2_O decomposition, CO_2_ conversion, and important approaches to enhance photocatalytic activity are highlighted. Photocatalytic enhancement via the surface plasmon resonance effect is also examined and the photocatalytic performance of the bismuth-based photocatalytic mechanism is elucidated and discussed in detail, considering the different semiconductor junctions. Finally, the current challenges and future development of bismuth-based photocatalysts are described.

## 2. Background

Bismuth(III) oxide (Bi_2_O_3_) exists in six crystallographic polymorphs; namely, monoclinic *α*, tetragonal *β*, body-centered cubic *γ*, face-centered *δ*, orthorhombic *ε*, and triclinic *ω* [9]. Among them, the *α*, *β*, and *δ* phases show photocatalytic reactivity upon visible light irradiation [10]. Bi_2_O_3_ has been used as a heterogeneous photocatalyst capable of catalyzing the degradation of several synthetically important sunlight-driven pollutants. It possesses a narrow bandgap (2.1–2.8 eV) with useful photocatalytic activity. Due to the high-oxidation potential of valence band holes (+3.13 V vs. normal hydrogen electrode (NHE)), its photo-efficiency has been demonstrated in a variety of applications ranging from energy storage and pollutant degradation to bio-compound degradation [11,12,13]. Although Bi_2_O_3_ exhibits high efficiency in promoting photooxidation, its conduction band (CB) electrons (+0.33 V vs. NHE) are unable to interact with organic molecules because of the rapid recombination of charge carriers, which hinders its application in reduction processes. However, several studies have shown that the Bi_2_O_3_ photocatalytic activity can be improved either by doping or by combining two or more materials or tuning surfaces [14,15,16].

Recently, Bi_2_O_3_ has also become a popular photocatalyst for driving the photodegradation of organic dyes. Its photocatalytic activity was investigated for the formation of C-C and C-S bonds [17,18] and atom-transfer radical-addition-type reactions [19]. Bi_2_O_3_ is low-cost and non-toxic. Other advantages are its visible-light drive and high availability. Moreover, in some cases, it can replace the use of organo-metal complexes such as the photoredox catalysts Ru(bpy)_3_Cl_2_-combining 2,2′-bipyridine and expensive and not-abundant ruthenium [20]. The semi-metallic nature of Bi^0^ enables its use as a semiconductor photocatalyst, or as a cocatalyst to tune the photocatalytic behavior of the host material [21,22]. Bismuth is a metal element in group V B in the periodic table (*M*_w_ = 208.98 g mol^−1^) with the 6s^2^6p^3^ electron configuration [23]. The lone-pair distortion of Bi 6s orbitals in Bi-based composite oxides may lead to the overlap of O 2p and Bi 6s orbitals in the valence band, which is beneficial to reduce the bandgap and the mobility of photogenerated charges and improve the photoactivity [24]. At the same time, when the 6s orbital is empty, the Bi^5+^ valence state also has good absorption of visible light [25]. Bismuth exists as Bi^3+^ in most common Bi-type photocatalysts, such as complex oxides (BiVO_4_, Bi_2_WO_6_, BiPO_4_) [26,27,28], sulfides (Bi_2_S_3_) [29], and oxyhalides (BiOI, BiOBr, BiOCl) [30,31,32,33]. It is therefore not surprising that a 14-fold increase in the number of reports related to Bi-based photocatalysts has been observed from 2010 to 2022 [34,35].

In order to improve the mineralization rate of organic dyes in wastewater, high charge separation efficiency, long-term stability, suitable band edge positions, and good redox capacity are also required for high-efficiency photocatalysts in addition to suitable bandgaps. Therefore, due to the unsatisfactory photocatalytic activity of single-component photocatalysts, various controllable bismuth-based compounds have been synthesized through morphological structure mediation, the construction of heterostructures or nanostructures, the doping of metal elements, and defect site mediation [36,37,38,39]. In particular, S,F-codoped Bi_2_WO_6_ with oxygen vacancies synthesized via hydrothermal calcination and post-sulfurization showed a photocatalytic performance in Cr(VI) reduction of 94.3% and methyl orange degradation of 95.4% in 120 min under visible light [40]. Among defects, oxygen vacancies (OVs) have been shown to improve photocatalytic activity [41,42]. However, the surface oxygen defects are unstable due to easy oxidation during the photocatalytic reaction process [43]. An exception is provided by Bi_5_O_7_Br nanotubes synthesized with OVs by combing water-assisted self-assembly and a low-temperature wet chemical approach [44]. In this case, the surface OVs were stable and able to capture and activate N_2_, reducing it to NH_3_, in pure water. The properties of many Bi-based photocatalysts and their performance in organic dye degradation and H_2_ production are described in detail in the following sections.

### 2.1. Fundamental Mechanism and Main Active Species of Bismuth-Based Photocatalysts

The main mechanism of bismuth-based photocatalysts can be summarized as photon absorption, excitation, and reaction processes. The photocatalytic process can be applied not only to the degradation of dyes but also to the degradation of antibiotics. Specifically, the photocatalytic degradation of antibiotics based on bismuth-based semiconductors is an effective, eco-friendly, and promising method for toxic substances. The predominant mechanisms of the bismuth-based photocatalysts for antibiotic photocatalytic degradation could be summarized as absorbing photons, excitation, and reaction, as shown in Figure 1. The antibiotics, as well as their intermediates, are converted to small-molecule compounds via the oxidation of oxygen species (h^+^, •O_2_^−^ or •OH) and eventually decomposed into CO_2_ and H_2_O [45].

In detail, when a photocatalyst absorbs photons with energy higher than its bandgap, the valence band (VB) electrons can be excited and jump into the CB. A photohole is inseparable from a photocatalyst and is a vacancy in its crystal lattice. Thus, the photocatalytic process is expressed as follows:photocatalyst + hv → (photocatalyst + h^+^) + e^−^,(1)
then, the photogenerated electrons and holes are effectively separated and migrated to the photocatalyst surface. The photo-induced holes directly attack dye molecules, as follows:h^+^ + dye → H_2_O + CO_2_ + degradation product,(2)
theoretically leading to the significant degradation of these pollutants. Furthermore, when the holes further migrate to the photocatalyst surface, the oxidation pathway starts, accompanied by the oxidation of H_2_O/OH^−^ to generate hydroxyl radicals (•OH), such as:H_2_O/OH^−^ + h^+^ → •OH + H^+^.(3)

Meanwhile, the typical redox potential of the photocatalyst should be higher than •OH/OH^−^ (+1.99 eV). Furthermore, hydroxyl radicals have stronger oxidation potential (*E*^0^ = 2.8 eV) and lower selectivity than other oxidants during the decomposition of water pollutants [46]. Remarkably, the top of the VB of most Bi-based catalysts is higher than the redox potential of •OH/OH^−^, indicating that hydroxyl radicals are easily generated during Bi-based catalysis. In fact, the reaction pathway between hydroxyl radicals and dye molecules can be summarized as follows: (i) •OH and dye molecules simultaneously adsorb on the catalyst surface and then react spontaneously, (ii) •OH in aqueous solution and adsorbed on the photocatalyst surface reacts with dye molecules, (iii) •OH adsorbed on the catalyst surface reacts with the surrounding dye molecules, and (iv) finally, •OH reacts with the dye molecules in the aqueous solution. Generally, these main pathways are considered for the degradation of dye molecules by bismuth-based photocatalysts [47]. Figure 2 displays the band edge positions of bismuth-based photocatalysts.

Conversely, if the CB potential of the semiconductor is negative compared to the O_2_/•O_2_^−^ redox potential (−0.13 V vs. NHE), a reduction pathway can also be observed, in which O_2_ is reduced by electrons to •O_2_^−^ (O_2_ + e^−^ → •O_2_^−^). The excited H_2_ ions will recombine with electrons and generate thermal energy (H^+^ + e^−^ → energy), which reduces the photodegradation efficiency of the catalyst. Then, dye molecules and their intermediates are converted into small molecular compounds through the oxidation of O_2_ species (h^+^, •O_2_^−^ or •OH) and finally decomposed into CO_2_ and H_2_O (dye molecules + radicals (•OH or •O_2_^−^) → CO_2_ + H_2_O + small molecule compound). In contrast, due to their different electronic structures, the effect of various active substances on the degradation of dye molecules differs. In order to direct the preferential active species during the reaction, different scavengers such as MeOH (for •OH), KI (for h^+^), *p*-benzoquinone (for •O_2_^−^ radicals), and AgNO_3_ (for e^−^) were introduced into the reactor to trap the active species [48]. Apparently, besides superoxide radical (•O_2_^−^), hydrogen peroxide (H_2_O_2_) and hole (h^+^) play a major role in the photodegradation process of most organic, dye-based, bismuth-based photocatalysts [49]. For example, the main reason for their higher photocatalytic degradation may be that the dye molecules are considered vulnerable to h^+^ attack [50]. In addition, hydroxyl radicals (•OH) typically react rapidly and non-selectively with most organic pollutants, so they also play a key role in the degradation of dye molecules. It is worth noting that the resulting •O_2_^−^ is unstable and prone to disproportionation reaction to generate other reactive oxygen species including •OH.

### 2.2. Significance of Nanostructure or Heterostructure or Heterojunction or Nanointerfaces

In fact, a practical approach to improve the photo-response in photoactive materials is the formation of nanointerfaces (heterojunctions) by coupling with other semiconducting materials or metals. The formation of heterojunctions has been widely used in visible-light-responsive photocatalytic dye degradation and H_2_O splitting in batteries [51,52,53]. It is also worth noting that heterojunctions in nanomaterials can be mainly classified into three different types: (1) type I is a straddling gap, where the VB and CB energies of the cocatalyst are higher and lower than those of the photocatalyst; (2) type II is a staggered gap, the VB and CB of the cocatalyst are higher than that of the catalyst; (3) Z-scheme has the same band structure as Type-II, but with different charges of the acceptor/donor pair carrier transfer pathway, which may enhance redox capacity. The Z-scheme configuration can further be differentiated into three types; namely, direct Z-scheme (mediator free), solid-mediator, and redox pair mediator types (see Figure 3 [54]).

Indeed, the band gap and Fermi level can be tuned at their interfaces, providing charge separation and facilitating alternative paths for excited electrons to prevent charge recombination. For example, Shan and co-workers [55] proposed a band alignment of α-Bi_2_O_3_/BiOCl (001) core-shell heterojunctions based on the shifted positions of the CB and VB to facilitate the accumulation of photoinduced electrons at their interfaces. Volnistem et al. [56] constructed mechanistically synthesized BiFeO_3_/Fe_3_O_4_ nanostructures and used them to degrade MB dyes under visible light. This study shows that the nanointerface promotes the ferrous Fe^2+^ ions of Fe_3_O_4_ to enhance the catalytic efficiency compared to the bulk. The direct Fenton-like method is another effective method to degrade dyes using Fe^2+^ ions and H_2_O_2_. The presence of Fe^2+^ ions combined with the photo-Fenton process can enhance the decomposition of H_2_O_2_ into oxidative radicals, thereby increasing the degradation rate. In another study, Liu et al. [57] reported that the hydrothermally synthesized Bi_4_Ti_3_O_12_/BiOI nanostructures degrade BPA. The results showed that the degradation was 12 times faster than that of Bi_4_Ti_3_O_12_ crystallites, which was attributed to the internal electric field in the ferroelectric domains under the external electric field. The improved internal electric field in ferroelectric catalysts can facilitate the separation and transfer of charge carriers, driving more carriers to the surface of the photocatalytic material, thereby enhancing its photocatalytic efficiency.

BiO*X*/CuFe_2_O_4_ (*X* = Br, Cl, and I) nanostructured p–n junctions were constructed by hydrothermal and coprecipitation methods [58]. Such a nanostructure induces a built-in electric field at its interface, which facilitates the transfer of pattern changes, indicating a significantly enhanced visible-light-driven photoactivity without the use of any cocatalyst. The BiOI or BiOBr/CuFe_2_O_4_ nanostructure demonstrates the conventional type I and type II charge-transfer mechanism, which can effectively reduce the charge-transfer resistance compared with the bulk structure. Importantly, the direct Z-type mechanism of BiOCl/CuFe_2_O_4_ nanostructures has formed tight interfacial contacts, resulting in a 5.7-fold increase in H_2_ release compared to bare BiOCl and improved catalytic efficiency by a factor of two compared to type II BiOI/CuFe_2_O_4_ nanostructures. The study also shows that the low resistance of the Nyquist plot confirms the superiority of the direct Z-scheme in promoting charge separation and transfer and increasing carrier density. Furthermore, by designing the band-edge potential, the BiO*X*/CuFe_2_O_4_ heterostructure achieves optimal space charge layer width and redox potential, which reduces the fast recombination rate. This work delivers a model for designing highly engineered BiO*X*-based nanostructures with tuned band edges for efficient photocatalytic activity. Therefore, the construction of nanostructures or heterostructures is of great significance for improving photocatalytic efficiency.

### 2.3. Surface Plasmon Resonance Effects in Bismuth-Based Photocatalysts

The surface plasmon resonance (SPR) effect of noble metals such as gold and silver is currently used to enhance the visible photocatalytic activity of semiconductor photocatalysts [59]. This mechanism is attributed to the huge local electric field enhancement observed at the surface of metallic nanoparticles (NPs) due to the interaction with the electric and magnetic fields of light. The excitation of electron-hole (e^−^-h^+^) pairs is boosted in the catalyst with the enhanced near-field of NPs, improving the photocatalytic activity. Although, the SPR effect of Bi nanospheres has been used to stimulate the excitation of photo-generated e^−^-h^+^ pairs in Bi-based semiconductor photocatalysts by the deposition of Bi on their surface. Improvement via the SPR effect of visible photoreactivity has been demonstrated for BiOBr [60], Bi_4_MoO_9_ [61], BiPO_4_ [62], and Bi_2_WO_6_ [63].

## 3. Synthesis Strategies of Bismuth-Based Photocatalysts

Various strategies have been proposed for the preparation of bismuth-based photocatalysts with desirable structure and morphology [64]. Synthesis methods of bismuth-based photocatalysts include room-temperature solid-state milling [65], high-temperature solid-state reaction [66], the precipitation method [67], the hydro/solvothermal technique [68,69], the ion-exchange route [70], the microwave-assisted method [71], and the microemulsion-based route [72]. Lu et al. [73] reported the synthesis of β-Bi_2_O_3_ microrods of ~1 µm in width by a solution crystallization technique at 70 °C, without further calcination treatment. The metastable tetragonal β-Bi_2_O_3_ crystalline powders with orange color transformed into yellow monoclinic α-Bi_2_O_3_ crystals after 60 min of reaction. The morphology of a pumpkin was changed from the microrod-like structure of β-Bi_2_O_3_ crystals to the large rhomboid structure of α-Bi_2_O_3_ crystals. This synthesis method reflects the size control process of metastable β-Bi_2_O_3_. By simply adjusting the experimental parameters such as NaOH concentration, stirring, and reaction temperature, metastable β-Bi_2_O_3_ crystals can be stably stored in the reaction system for different lengths of time. The pumpkin β-Bi_2_O_3_ nanostructures exhibited good photocatalytic performance for the degradation of RhB dyes under visible light irradiation.

### 3.1. Synthesis Strategies of Bismuth Oxides

One-dimensional (1D) Bi_2_O_3_, including nanotubes [74], nanowires [75], nanosheets [76], and nanorods [77], holds promise for photocatalytic activity. Tien et al. [78] reported the synthesis of α-Bi_2_O_3_ nanowires with a diameter of 500 nm and a length of up to 20 μm by catalyst-driven gas-phase transport, and the growth direction was (010). The growth mechanism of α-Bi_2_O_3_ nanowires is explained as a two-step growth model, which considers the formation of crystal planes catalyzed by gold and the growth of α-Bi_2_O_3_ nanowires during bismuth catalysis. It is revealed that the formation and growth mechanism of α-Bi_2_O_3_ nanowires is influenced by Au nanoparticles. In detail, the formation of nanowires is shown in Figure 4. In steps 1 and 2, the Au-catalyzed growth of the precursor vapor is adsorbed onto Au nanoparticles, where facets are formed between the Au/Bi interface under an oxidizing environment. Once a facet is formed, it can serve as a nucleation site for further one-dimensional growth. At this stage, growth has shifted into different growth patterns (steps 3 and 4). Since the growth occurs at the dual-surface interface, the growth is driven by a dual catalytic mechanism. By comparing nucleation on heterogeneous solid surfaces (gold nanoparticles) and self-nucleation (sapphire surfaces), nucleation on gold catalysts interacting with nuclei will have lower free energy than self-nucleation. Therefore, the degree of supersaturation required for self-nucleation is much higher than for heterogeneous nucleation. Finally, the formation of crystal planes provides the nucleation and growth of α-Bi_2_O_3_ nanowires, which facilitates a simple strategy to control the nucleation and structural characteristics of α-Bi_2_O_3_ nanowires.

Xiao et al. [79] reported the solvothermal synthesis of β-Bi_2_O_3_ nanospheres followed by a calcination process. In detail, monodispersed bismuth nanospheres were formed by a solvothermal process with D-fructose as the main reducing agent, followed by calcination in air to transform into β-Bi_2_O_3_ nanostructures, revealing that the D-fructose concentration significantly affects the structural β-characteristics of Bi_2_O_3_ nanospheres. The growth mechanism of β-Bi_2_O_3_ nanospheres involves the in situ reduction of Bi(III)-ethylene glycol complex spheres as self-sacrificial agents, followed by the in situ oxidation of bismuth nanospheres by contact with oxygen. The β-Bi_2_O_3_ nanospheres exhibited APAP degradation efficiency that was 79 times higher than that of TiO_2_ powder (Degussa P25), which was attributed to the suitable energy band structure, high oxidation potential, and good dispersion of β-Bi_2_O_3_ nanospheres. Higher photoactivity is supported by the experimental determination of reactive oxygen species during photocatalysis. However, secondary pollutants may still occur during the photocatalytic process, requiring an in-depth analysis of the treated water.

### 3.2. Synthesis Strategies of Bismuth Ferrites

BiFeO_3_, with a rhombohedral twisted perovskite structure and a narrow bandgap visible light response of 2.2 eV, is an attractive candidate for its fascinating application in novel photocatalyst materials [80]. BiFeO_3_ is a ferroelectric material, and the intrinsic internal electric field reduces the recombination of photoinduced charge carriers and increases the degradation rate. The low decomposition temperature of bismuth salts and the change in ionic valence make it difficult to prepare impurity-free BiFeO_3_ by conventional solid-state reactions at the evaluated temperature. Other strategies such as the soft sol-gel method, co-precipitation method, and solvent/hydrothermal method have been identified as promising strategies to prepare BiFeO_3_ nanoparticles with desirable morphology [81,82,83]. The hydrothermal synthesis of BiFeO_3_ without high-temperature calcination can better control the purity and morphology of the material by controlling the reaction conditions. Due to the influence of the number of reaction sites and the size of the bandgap energy, the shape and size control of the particles plays a very important role in the photocatalytic activity. Many different morphologies of BiFeO_3_ particles have been reported, including wires, tubes, submicron spindles, and rod-like particles [84,85], including synthetic steps and controlled processes of self-assembly or organization of BiFeO_3_ with regular geometry. However, the continuous regulation of the shape or size of microscale BiFeO_3_ and its effect on photocatalytic activity is an important topic of core research, which is of great significance for understanding the catalytic mechanism and developing dual-semiconductor photocatalysts.

Ferrite bismuth materials, including perovskite (BiFeO_3_), mullite (Bi_2_Fe_4_O_9_), and sillenite (Bi_25_FeO_40_), exhibit outstanding magnetic, electronic, and dielectric properties. Among them, mullite-based Bi_2_Fe_4_O_9_ is a competitive candidate photocatalyst to drive visible light-catalyzed oxidation reactions due to the band gap energy of 1.9–2.1 eV, with standard multi-band semiconductor properties [86]. However, the catalytic efficiency of Bi_2_Fe_4_O_9_ is relatively low due to the fast recombination of photogenerated electron-hole pairs [87]. The photosensitivity can be improved by the separation of electrons and holes in Bi_2_Fe_4_O_9_ with silver halides (Ag*X*, *X* = Br, I, and Cl). Unfortunately, the strong photosensitivity of silver halides leads to the reduction of Ag^+^ to Ag^0^ under light irradiation, which reduces their stability and lifetime, thus limiting their photocatalytic applications. In general, AgBr is a popular high-efficiency photocatalyst with a band gap of 2.6 eV. Ma et al. [88] developed a one-dimensional magnetically separable Bi_2_Fe_4_O_9_/C@AgBr nanostructured photocatalyst that can degrade 97.4% of MB within 60 min. The high catalytic performance is mainly due to the efficient charge separation and migration in the Bi_2_Fe_4_O_9_/C@AgBr nanostructures. In addition, carbon also promotes the chemical protection of nanostructures and improves the conductivity and stability of catalysts.

## 4. Recent Developments in Bi-Based Photocatalysts

### 4.1. Bi-Oxide Nanostructures

Bandgaps of Bi_2_O_3_ polymorphs range in the order of δ-Bi_2_O_3_ (3.0 eV) > α-Bi_2_O_3_ (2.8 eV) > β-Bi_2_O_3_ (2.1 eV) > γ-Bi_2_O_3_ (1.64 eV) [89]. Since γ-Bi_2_O_3_ has a narrow band gap, it can efficiently utilize light in the visible region of the solar spectrum. However, the photocatalytic efficiency of bare Bi_2_O_3_ is still unsatisfactory for dye degradation unless it is integrated or doped with other semiconducting compounds, especially for the δ-phase because of its wide bandgap value. In particular, the α-Bi_2_O_3_ type is a stable phase over a wide temperature range, while β-, γ-, and δ-Bi_2_O_3_ are metastable at 25 °C. For this reason, many strategies have been applied to enhance the photoactivity of Bi_2_O_3_. Barno et al. [90] pioneered the hydrothermally prepared heterojunction features of BiVO_4_/MnV_2_O_6_ photocatalysts for the photodegradation of RhB and MB dyes, showing that the BiVO_4_/MnV_2_O_6_ heterojunction photocatalyst achieved a degradation rate of 96% in 35 min and MB dye degradation efficiency reached 98% within 6 min under visible light exposure. This study shows that superoxide anion radical is the main responding species during dye degradation. Furthermore, the BiVO_4_/MnV_2_O_6_ heterojunction photocatalyst exhibits excellent 4-nitrophenol reduction in the presence of NaBH_4_, and 4-aminophenol is produced without intermediate by-products, thanks to the heterojunction properties and its suitable band alignment. By controlled hydrothermal synthesis, allowing the control of the ratio of {010} to {110} facets on BiVO_4_, which respectively serve as reductive and oxidative sites, Guan et al. obtained decahedral BiVO_4_ single crystals with superior photocatalytic water oxidation achieving efficient water splitting [91]. Tian et al. [92] reported thermochemically prepared β-Bi_2_O_3_/Mn_3_O_4_ nanostructures for the photodegradation of RhB, BPA, and MB and the removal of nitric oxide (NO). The optimized β-Bi_2_O_3_/Mn_3_O_4_-2 wt.% photocatalyst exhibits excellent photocatalytic activity for pollutant (RhB, MB, and BPA) degradation and NO removal. This efficiency was ascribed to tight contacts between the β-Bi_2_O_3_ and Mn_3_O_4_ at their interface, which possesses a type-II heterojunction photocatalytic mechanism. This mechanism facilitates the rapid separation of photo-induced charge carriers, resulting in excellent photocatalytic activity.

He et al. [93] pioneered solvothermally synthesized 3D flower-like β-Bi_2_O_3_/Bi_12_O_17_Cl_2_ nanostructures for the degradation of PTBP under visible light. The nanostructures are formed by reducing Bi(III) to nano-metallic bismuth, followed by the thermal treatment of bismuth with oxygen and bismuth oxide chloride hydroxide in the presence of air. The synthesized β-Bi_2_O_3_/Bi_12_O_17_Cl_2_ nanostructures have a good energy band structure, and a close-contact heterojunction is formed between the synthesized β-Bi_2_O_3_ and Bi_12_O_17_Cl_2_, with a high specific surface area and a hierarchical micro-nanostructure, thereby decomposing PTBP under visible light with excellent photo-mineralization efficiency. Compared with the as-synthesized Bi_12_O_17_Cl_2_, the optimally synthesized β-Bi_2_O_3_/Bi_12_O_17_Cl_2_ nanostructure exhibited 12-fold higher photocatalytic activity, which was attributed to the direct hole and superoxide radical oxidation rather than oxidation by hydroxyl radicals. Due to the presence of heterojunction features, the visible light absorption range is enhanced at the origin of their remarkable photoactivity under visible light illumination. Sun et al. [94] reported water thermal synthesis of α-/γ-Bi_2_O_3_ nanostructures for RhB degradation under visible light. The key parameters of the hydrothermal process are holding time, temperature, additive dosage, and pH conditions. Compared with α-Bi_2_O_3_ and γ-Bi_2_O_3_ nanostructures, α-/γ-Bi_2_O_3_ nanostructures exhibited higher RhB photodegradation activity, which was attributed to the synergistic effect of the homojunction. Gardy et al. [95] reported a solid reactive heat treatment made of α-/β-Bi_2_O_3_ nanopowders to degrade a mixed dye of RhB and IC under UV and visible light irradiation. It can be observed that the α-/β-Bi_2_O_3_ mixed phase produced 20% β-Bi_2_O_3_ phase after annealing at 550 °C, while the α-Bi_2_O_3_ heterojunction was formed after annealing at 650 °C. α-/β-Bi_2_O_3_ photocatalysts exhibit better efficient charge separation and activity via α-/β-Bi_2_O_3_ transfer, indicating that α-/β-Bi_2_O_3_ heterojunctions are more efficient than commercial α- and β-Bi_2_O_3_ materials separately.

### 4.2. Bismuth Vanadate (BiVO_4_)

BiVO_4_ has attracted extensive attention due to its remarkable structural, optical, and chemical properties, photocorrosion resistance, and good activity for the photocatalytic degradation of organic pollutants. The crystal structures of BiVO_4_ are monoclinic, orthorhombic, and tetragonal, among which the monoclinic with a band gap of about 2.4 eV has good photocatalytic activity compared with the other two forms. The phase transition from tetragonal to monoclinic occurs irreversibly at 500 °C. The basic building blocks are developed from the VO_4_ tetrahedron and the BiO_8_ dodecahedron. In addition, Bi and V atoms are alternately arranged along the crystallographic axis, which makes monoclinic BiVO_4_ exhibit the properties of a layered structure. However, BiVO_4_ has limited use as a photocatalyst because of its fast recombination rate of photogenerated carriers due to its band edge position. Furthermore, the photocatalytic efficiency of BiVO_4_ is much lower than expected due to its lower surface area and lower carrier separation and transfer ability. Therefore, topography control, cocatalysts and selective deposition, and coupling to other semiconductors to build nanostructures are needed. For example, Liu et al. [96] developed an in situ transformation of as-prepared BiVO_4_ with the help of NaOH to form BiVO_4_/Bi_25_VO_40_ nanostructures through a dissolution–recrystallization process, in which the monoclinic decahedron of BiVO_4_ was first etched with an alkaline solution on the preferential crystal planes (010) and converted to cubic Bi_25_VO_40_. In this study, the authors successfully controlled the concentration conditions of the alkaline solution to precisely tune the phase composition of the heterojunction by reducing BiVO_4_ and increasing Bi_25_VO_40_ in the nanostructures. Advancing from the in situ switching strategy and combined band structure, the type II tight heterojunction formed tight interfacial contacts. This led to fast charge transfer with the spatial separation of carriers, which considerably enhanced the photocatalytic degradation of tetracycline hydrochloride (TCHC) under visible light. In the obtained BiVO_4_/Bi_25_VO_40_ nanostructures, the role of Bi_25_VO_40_ is crucial in the photoactivity; the atypical bismuth-rich phase bismuth vanadate consists of the same elements as BiVO_4_ with a narrower bandgap of 2.1 eV, resulting in a wider range of visible light absorption. Since BiVO_4_ and Bi_25_VO_40_ have suitable energy band positions and approximate crystal structures, the rational coupling of BiVO_4_ and Bi_25_VO_40_ through an in situ synthesis process has been shown to yield close-contact heterojunctions with suitable band energies, leading to an enhanced charge-carrier transmission rate. In another study, Duan et al. [97] reported hydrothermally synthesized BiVO_4_/rGO nanostructures with the assistance of ethylenediaminetetraacetic acid disodium salt instead of nitrate, which facilitates the formation of a fully acidic environment to prevent the hydrolysis of Bi^3+^, and applied them as a photocatalyst for the degradation of RhB. The BiVO_4_/rGO nanostructured photocatalyst exhibited 98.3% degradation in 180 min under visible light. When photons land on the BiVO_4_/rGO surface, the electrons in the VB of BiVO_4_ are excited to the CB by leaving a hole on the VB. Since rGO acts as an electron acceptor with good electrical conductivity, the photogenerated electrons in the CB can move to rGO, which speeds up the separation efficiency and thus enhances photoactivity.

El-Hakam et al. [98] reported the ultrasound-assisted introduction of mesoporous SiO_2_ (i.e., *m*-SiO_2_) on BiVO_4_ nanoparticles to control the size of BiVO_4_ nanoparticles to 2.4–5.1 nm on m-SiO_2_ to form BiVO_4_/m-SiO_2_ nanostructures. This BiVO_4_/*m*-SiO_2_ nanostructure was used to degrade MB and BG dyes as a function of *m*-SiO_2_ in the nanostructure. Compared with bare BiVO_4_, the BiVO_4_/m-SiO_2_ nanostructured photocatalyst exhibits remarkable photoactivity, which is attributed to the synergistic effect between *m*-SiO_2_ and BiVO_4_, which enhances the separation of charge carriers. The effects of operating parameters such as dye concentration, *m*-SiO_2_ content, reaction time, and temperature were closely related to the photocatalytic activity. The nanostructure with a 10 wt.% *m*-SiO_2_/BiVO_4_ sample exhibited the highest photocatalytic activity. Since the silica in the nanostructure is in close contact with the BiVO_4_ nanoparticles, the photoelectron conversion of BiVO_4_ is improved by reducing the recombination charge carriers based on the suitable band positions of BiVO_4_, which is found to be reusable.

### 4.3. Silver-Bismuth Photocatalysts

Advanced oxidation methods are considered promising methods to solve environmental problems by releasing free radicals, which have strong oxidative power. Furthermore, chemical oxidation and photocatalysis are two common tools for removing pollutants from wastewater. Compared with semiconducting oxides, perovskite-group silver-bismuth-based photocatalysts have different crystal structures, which provide a wide range of degrees for tuning their physicochemical properties. Recent studies have shown that silver bismuth, through defect engineering, can effectively enhance its photoactivity due to its physical adsorption and chemical oxidation [99]. Therefore, it significantly improves the mineralization ability of organic dyes. Typically, silver bismuth is synthesized from AgNO_3_, which replaces the Na ions of NaBiO_3_ in a hydrothermal reaction. The silver-bismuth photocatalyst has chemical oxidation abilities mainly due to the release of lattice oxygen in bismuthate, which is partially converted into active oxygen. When these reactive oxygen species come into contact with dye molecules, large amounts of reactive oxygen species are released. However, due to the irreversible transformation of lattice oxygen into chemisorbed oxygen, accompanied by the transformation from Bi(V) to Bi(III), the photocatalytic performance of silver bismuth single compounds decreases to varying degrees [100]. Many studies have been carried out to promote the release of reactive oxygen species, thereby enhancing the catalytic ability of silver bismuth. Silver bismuth was converted into α-/β-Bi_2_O_3_/Ag_2_O nanostructures by a simple calcination process, during which the morphology was transformed from nanosheets to porous nanosheets and ravines, for the degradation of TC under visible light [99]. After the deactivation of silver bismuth, the lattice oxygen is transformed into chemically and physically adsorbed oxygen, and numerous carbon species can be adsorbed onto the surface of the material. Bi species in silver bismuth are transformed to β-Bi_2_O_3_, and all Ag species are converted to AgO_2_. Furthermore, with increasing temperature, the β-Bi_2_O_3_ phase transforms into α-Bi_2_O_3_ identified by the color of the sample, and morphological changes may occur as described above. β-Bi_2_O_3_/Ag_2_O activated at 290 °C exhibited the best degradation efficiency of 78% within 2 h, and its reaction rate constant was 3.7 times higher than that of silver bismuth due to the low recombination probability and strong photoresponsivity. Figure 5 shows the possible photocatalytic mechanism of silver-bismuth-based photocatalysts.

### 4.4. Bismuth Oxide Silicate-Based Photocatalysts

The compounds with molecular formula Bi_2_*X*O_20_ (*X* = Si, Ti, Ge, Pb, etc.) are called bismuth sillenites and are considered promising materials for developing low-temperature co-fired ceramic technology [101]. Mainly bismuth silicates, such as Bi_2_SiO_5_, Bi_12_SiO_20_, and Bi_4_Si_3_O_12_, have received increasing attention, in particular Bi_2_SiO_5_, as an alternative to conventional lead-based ferroelectric materials with a phase transition temperature of 673 K. Bi_2_SiO_5_ crystallizes with an orthorhombic structure (space group *Cmc*2_1_) with lattice constants *a* = 15.19 Å, *b* = 5.68 Å, *c* = 5.314 Å and *Z* = 4 [102]. As the general formula of the Aurivillius-like structure is (Bi_2_O_2_)[A*_m_*_−1_(B)*_m_*O_3_*_m_*_+1_], Bi_2_SiO_5_ with *m* = 1 is composed of BiO_4_ pyramids in the [Bi_2_O_2_]^2+^ layers and [SiO_3_]^2−^ layers, as shown in Figure 6a [103]. First-principle calculations suggest that the polarization of Bi_2_SiO_5_ originates from the SiO_3_ layer rather than the Bi_2_O_2_ layer [104].

Figure 6b–e confirm the concept of fragments for the experimental extraction of single dipole units derived from BiO and SiO_3_ clusters. The boundaries of a segment can be determined by local minima around the segment. Therefore, the fragments satisfy the charge neutrality of Bi_2_SiO_5_, and their partial electrical polarization is estimated by considering the volume of the unit cells. Bi_2_SiO_5_ crystals have special properties, namely dielectric properties, thermoelectric properties, and nonlinear optical properties, and have ferroelectric properties due to their non-centrosymmetric structure, and the band gap of 3.54 eV is narrower than that of BiPO_4_ (3.85 eV) [105]. Actually, the inductive effect of PO_4_^3−^ benefits the separation of the photoinduced electron-hole pairs, but the large energy gap of BiPO_4_ implies that this compound is a good photocatalyst only in UV light, which accounts for 4% of solar irradiation. Therefore, the formation of nanostructures with a smaller gap material such as Bi_2_SiO_5_ is a means of avoiding this drawback.

Co-precipitated hydrothermally synthesized Bi_2_SiO_5_/BiPO_4_ nanostructures were constructed and applied to degrade phenol and MB dyes under UV-light irradiation [106]. This work revealed the extension of the photoresponse range of BiPO_4_ by coupling with Bi_2_SiO_5_ and forming a type-II heterojunction. Bi_2_SiO_5_/BiPO_4_ nanostructures exhibit significantly enhanced photoactivity against phenols and dyes, being 4.36-fold and 1.13-fold higher with respect to Bi_2_SiO_5_. This improvement was attributed to the significantly enhanced charge separation ability, expanded absorbance, and good crystallinity through the heterojunction. A synergistic effect was observed with medium Brunauer–Emmett–Teller specific surface area. The energy (vs. NHE) of the bottom of the CB of BiPO_4_ (−0.65 eV) is more negative than that of Bi_2_SiO_5_ (0.05 eV), while the top of the VB of BiPO_4_ is at 3.2 eV against 3.59 eV for Bi_2_SiO_5_. Therefore, the two components, BiPO_4_ and BiSiO_5_, have matched charge potentials, which can facilitate the flow of charge carriers through their interfaces. Furthermore, it is reasonable to design a type-II heterojunction as a novel and robust photocatalytically active system by coupling BiPO_4_ with a narrower bandgap semiconductor, Bi_2_SiO_5_. Zou et al. [107] reported a one-pot solvothermal synthesis of Bi_2_SiO_5_/Bi_4_MoO_9_ nanostructures for the degradation of CIP under UV-light irradiation. The results show that the Bi_2_SiO_5_/Bi_4_MoO_9_ nanostructures exhibit higher photoactivity than Bi_2_SiO_5_ and Bi_4_MoO_9_; such a heterostructure not only suppresses the recombination of photoexcited charge carriers but also enhances light absorption. In addition, the effects of initial CIP concentration and coexisting ions on the photodegradation process of Bi_2_SiO_5_/Bi_4_MoO_9_ nanostructures were also confirmed. Figure 7 pictures the density-functional theory results of Bi_2_SiO_5_ and Bi_4_MoO_9_, indicating that the VB and CB of Bi_2_SiO_5_ and Bi_4_MoO_9_ are the same *k*-space, indicating that the intrinsic optical transition properties of Bi_2_SiO_5_ and Bi_4_MoO_9_ are direct transitions. The estimated theoretical band gaps of Bi_2_SiO_5_ and Bi_4_MoO_9_ are 3.69 and 2.86 eV, respectively, which are in good agreement with the experimental results. The total and partial electronic state densities of Bi_2_SiO_5_ and Bi_4_MoO_9_ are shown in Figure 7c,d. The VB top of Bi_2_SiO_5_ is mainly composed of O 2p and Si 3p orbitals, while the CB bottom is mainly composed of Bi 6p orbitals (Figure 7c). In the case of Bi_4_MoO_9_, the VB top is mainly composed of O 2p states, while the CB bottom is contributed by Mo 4d states.

In general, the sol-gel route has potential advantages over traditional solid-state synthesis methods because it allows precise control over composition, coating deposition, and uniformity. Veber et al. [108] proposed a synthetic procedure for bismuth silicate, consisting of bismuth nitrate pentahydrate, dried in a vacuum oven at 65 °C for 96 h to remove water contamination. The dehydrated bismuth nitrate was then dissolved in acetic acid and placed in a magnetic stirrer for 2 h. Another solution was prepared with tetraethoxysilane (Si(OC_2_H_5_)_4_, TEOS) and 2-ethoxyethanol with continuous stirring for 30 min. 2-ethoxyethanol can be used as a solvent for TEOS. After the two solutions were stirred separately, they were mixed and placed under magnetic stirring for 3 h and adjusted to pH 4 with ethanolamine. Di et al. [109] reported Bi_2_SiO_5_ nanosheets modified by carbon quantum dots (CQDs) with a diameter of 3 nm and applied them to the photocatalytic degradation of RhB under UV-light irradiation. CQDs-modified Bi_2_SiO_5_ nanosheets were shown to accelerate the charge transfer between the interiors of Bi_2_SiO_5_ nanosheets and promote the separation of surface charge carriers. The active species that enhance the photocatalytic activity are hydroxyl radicals and superoxide radicals, as evidenced by electron spin resonance analysis. Under UV-light irradiation, electrons are transferred from the VB to the CB of Bi_2_SiO_5_. Electrons migrating from the surface of Bi_2_SiO_5_ are transferred to the CQDs through the interface between the CQDs and Bi_2_SiO_5_. Electrons on the CQDs reduce the adsorbed O_2_ to •O_2_^−^, while holes on the VB oxidize OH^−^ to OH. The generated reactive oxygen species play a key role in the subsequent photocatalytic degradation process. Yang et al. [110] developed nanostructures of Bi_2_SiO_5/_g-C_3_N_4_ using a controllable hydrothermal method. The synthesized Bi_2_SiO_5/_g-C_3_N_4_ nanostructures were applied to the degradation of crystal violet (CV) dyes under visible-light irradiation, and the reaction rate constant was 0.1257 h^−1^, which was five times and three times higher than that of Bi_2_SiO_5_ and g-C_3_N_4_, respectively. From electron spin resonance and scavenger-test results, it was revealed that •O_2_^−^ active species played a major role in the degradation of CV dyes, while other primary reactive oxygen species such as •OH, h^+^ and ^1^O_2_ played a secondary role (where ^1^O_2_ is the first excited state of molecular oxygen (O_2_), known as singlet oxygen). Wu et al. [111] developed Bi_2_SiO_5_-SiO_2_, and Bi_12_SiO_20_-SiO_2_ photonic crystal films were prepared by spin-coating Bi_2_SiO_5_ or Bi_12_SiO_20_ on SiO_2_ photonic crystals and used as photocatalysts for the degradation of RhB dyes under UV-light irradiation. The photon localization of SiO_2_ photonic crystal plays a key role in improving the light absorption of bismuth silicate. This study provides a simple approach to improve the light-harvesting efficiency of photocatalysts and expand the application of photonic crystals. However, the thickness of the bismuth silicate film exhibits dual photocatalytic activity; with the thickening of bismuth silicate, the light absorption of bismuth silicate increases, but the photon localization weakens.

Building heterojunctions or nanostructures to prolong the lifetime of electron/hole pairs is a very important strategy to endow them with excellent photoactivity. However, developing nanostructures of bismuth silicate with the same composition, but forming different crystal structures, and with suitable band gaps, remains challenging. Jia et al. [112] developed different crystal structures using a one-pot hydrothermal synthesis method without the addition of other inorganic materials. However, the dose of cetyltrimethylammonium bromide (CTAB) is the key to modulating the formation of bismuth silicate crystal phases with assembled nanostructures and their surface states. When the concentration of CTAB was 1.5–2 mmol, Bi_2_SiO_5_ nanoparticles were anchored on Bi_12_SiO_20_ or Bi_4_Si_3_O_12_ nanosheets. The obtained two kinds of bismuth silicate nanostructures, Bi_2_SiO_5_/Bi_12_SiO_20_, have rod-like structures, and Bi_2_SiO_2_/Bi_4_Si_3_O_12_ have flower-like nanostructures. Owing to these two nanostructures, the optimized bismuth silicate material exhibits high photoactivity and remarkable cycling stability. Specifically, the degradation rate of RhB under visible light can reach as fast as 15 min with a reaction rate constant of 0.34 min^−1^, which is 189 times faster than other reports. This one-pot synthesis strategy for developing single-component nanostructures has significant implications for designing other novel photocatalysts based on their natural multivalent states, or various crystals such as Mn-, Fe-, and V-based nanostructures. Liu et al. [113] reported novel nanostructures of Bi_4_O_5_Br_2_/Bi_24_O_31_Br_10_/Bi_2_SiO_5_ developed by in situ ion exchange. The successful formation of nanostructures between bismuth bromide and Bi_2_SiO_5_ can be attributed to their structural similarity, thermodynamic tolerance, and high lattice matching. The novel nanostructured photocatalysts have well-aligned span bands at their closely contacted interfaces and exhibit remarkable photoactivity for phenol degradation under visible light. This ternary nanostructure exhibits about 2.5 times higher photoactivity against phenol than bulk BiOBr. The detailed photocatalytic mechanism of Bi_4_O_5_Br_2_/Bi_24_O_31_Br_10_/Bi_2_SiO_5_ nanostructures shows that in this ternary nanostructure (Figure 8), Bi_4_O_5_Br_2_ and Bi_24_O_31_Br_10_ have narrower band gaps than BiOBr, so they can absorb more long-wavelength light and improve the light utilization rate. For example, Bi_4_O_5_Br_2_ nanosheets with vertically aligned facets exhibited ∼6 times greater visible-light photodegradation efficiency against BPA than that of BiOBr nanosheets [114].

Based on the electrical structure, the band potentials of Bi_4_O_5_Br_2_, Bi_24_O_31_Br_10_, and Bi_2_SiO_5_ are compatible, forming a heterojunction with well-aligned cross-bands when they are in close contact. The photogenerated electrons in the CB of Bi_24_O_31_Br_10_ easily migrate to the CB of Bi_4_O_5_Br. Meanwhile, the holes formed in the VB of Bi_4_O_5_Br_2_ are easily transferred to the VB of Bi_24_O_31_Br_10_ and occur also in the VB of Bi_2_SiO_5_. Thus, long-lived reactive photo-charges can be generated, allowing for improved charge separation at their interfaces. The nanostructure-enhanced photocatalytic activity can be attributed to (i) improved photo-utilization due to the presence of multi-components with narrower band gaps, (ii) significantly improved charge separation capability due to the well-aligned cross-band structure, and (iii) having a large specific surface area due to their layered features, which can generate abundant active sites for catalytic reactions. This study may help to design novel nanostructured photocatalysts with higher photoactivity.

The photoreactivity of Bi/Bi_2_WO_6_ was found to steadily increase from 12.3% to 53.1% with increases in the number of Bi nanospheres from 0 to 10 wt% due to the SPR effect of Bi nanospheres on the Bi_2_WO_6_ photocatalyst [63]. After the modification of Bi_2_WO_6_ microspheres with Bi nanospheres, the photo-generated carriers can transfer from the CB of Bi_2_WO_6_ microspheres to Bi nanospheres, retarding the recombination. In addition, the near-field enhancement produced on Bi nanospheres by the SPR effect can significantly enhance the energy of electrons, which then consequently boosts the separation and migration of photo-generated carriers. Then, highly concentrated reactive oxygen species such as •O_2_^−^ and •OH radicals are produced in Bi/Bi_2_WO_6_ to oxidize NO (Figure 9).

Zhang et al. [115] developed Bi-induced Bi_2_O_2_CO_3_/Bi_12_SiO_20_ (i.e., Bi/Bi_2_O_2_CO_3_/Bi_12_SiO_20_) nanostructures grown in situ on Bi_2_O_2_CO_3_ on Bi_12_SiO_20_ by the oil bath method and applied them to the degradation of RhB and TC dyes. RhB and TC degradation by Bi/Bi_2_O_2_CO_3_/Bi_12_SiO_20_ nanostructures under simulated light are 12 times and 3.3 times higher than those of RhB and TC, respectively, which is attributed to the synergy effect of Bi heterojunction and the SPR effect. The trapping test results showed that •O_2_^−^ played a key role in the photodegradation process. The band gaps of Bi_2_O_2_CO_3_ and Bi_12_SiO_20_ materials are 2.57 and 3.2 eV, respectively. The estimated CB for Bi_12_SiO_20_ is −0.65 V vs. NHE, which is more negative compared to Bi_2_O_2_CO_3_ (−0.59 V vs. NHE). Electrons in Bi_12_SiO_20_ can migrate to the CB of Bi_2_O_2_CO_3_. Moreover, since the redox potential of •O_2_^−^ is −0.33 V with respect to NHE, the photoinduced e^−^ accumulation on the CB of Bi_2_O_2_CO_3_ further generates a large amount of ^•^O_2_^−^-degrading dyes. Meanwhile, the VB (2.61 eV) of Bi_2_O_2_CO_3_ is still more negative than that of Bi_12_SiO_20_, and the h^+^ on the VB of Bi_2_O_2_CO_3_ can be transferred to Bi_12_SiO_20_, thereby directly degrading the dye. Therefore, the nanostructure composed of Bi_12_SiO_20_ and Bi_2_O_2_CO_3_ effectively promotes carrier separation and enhances photoactivity. In addition, the SPR effect of Bi on the surface of Bi_2_O_2_CO_3_/Bi_12_SiO_20_ nanostructures not only broadens the light absorption and improves the light utilization efficiency, but also increases the surface electron excitation and interfacial electron transfer rate. The local electromagnetic field induced by the SPR effect of metallic Bi also promotes the migration and separation of charge carriers in the Bi_2_O_2_CO_3_/Bi_12_SiO_20_ nanostructures. The Fermi level of Bi (−0.17 eV vs. NHE) is more negative than the CB of Bi_12_SiO_20_, •O_2_^−^ can also be generated on Bi, and e^−^ from Bi_12_SiO_20_ can be moved to metallic Bi to facilitate carrier separation and enhance photoactivity. The synergy effect and SPR effect of Bi/Bi_2_O_2_CO_3_/Bi_12_SiO_20_ nanostructures are caused by metallic Bi, making Bi/Bi_2_O_2_CO_3_/Bi_12_SiO_20_ exhibit good photoactivity.

### 4.5. Bismuth and Bismuth-Rich Oxyhalides

Recently, a promising photocatalyst of the bismuth family, bismuth oxyhalide (BiO*X*, *X* = I, Cl, and Br), has been shown to induce more efficient charge separation due to its unique layered structure with an internal electrostatic field perpendicular to each layer, thus exhibiting significant photoactive performance [116]. Among them, BiOI has the smallest bandgap and strong absorption in the visible-light region. BiOI is a p-type semiconductor with a narrow bandgap of 1.8 eV, enabling it to absorb and utilize visible light. Therefore, it exhibits good photoactivity under sunlight exposure. Other forms of BiOI materials including the Bi_4_O_5_I_2_, Bi_7_O_9_I_3_, β-Bi_5_O_7_I, and α-Bi_5_O_7_I types have been widely reported [117]. However, the bandgap energy of these compounds is higher than that of BiOI, although lower than that of Bi_2_O_3_ [118]. Therefore, these materials are used as visible-light-induced photocatalysts. Interestingly, the structural and compositional features of bismuth iodide strongly affect its optical power, oxidative power, electronic properties, and other physicochemical properties, providing opportunities to obtain novel nanostructure photocatalysts for the efficient degradation of pollutants with different characteristics. Xiao et al. [119] reported that high-purity Bi_4_O_5_I_2_ with a hierarchical nanoflake structure can be easily obtained by reacting Bi^3+^, I^−^, and OH^−^ under solvothermal conditions at pH values of 6–10. The as-prepared Bi_4_O_5_I_2_ nanoflakes have a band gap of 2.17 eV, a CB edge potential more negative than the superoxide radical reduction potential, and a specific surface area of about 39 m^2^g^−1^. Bi_4_O_5_I_2_ nanoflakes exhibited excellent photoactivity and mineralization efficiency for the degradation of PTBP under visible light, and the reaction rate was 6.8 times higher than that of BiOI microspheres. More importantly, the as-prepared Bi_4_O_5_I_2_ nanoflakes remain stable during the photoreduction process and can be reused.

Bi_4_O_5_I_2_/Bi_4_O_5_I_2_ nanostructures were synthesized by the electrostatic self-assembly method and used to degrade RhB under visible light irradiation [120]. In detail, two different Bi_4_O_5_I_2_ compounds were synthesized using an ionic iodine source, namely [Hmin]I (1-hexyl-3-methylimidazolium iodide) and a KI source. The Bi_4_O_5_I_2_(KI) was negatively charged, while [Hmin]I was positively charged, resulting in an electrostatic attraction between Bi_4_O_5_I_2_(KI) and Bi_4_O_5_I_2_([Hmin]I) to form a final product in the form of Bi_4_O_5_I_2_([Hmin]I) nanosheets introduced in the Bi_4_O_5_I_2_(KI) bulk. This final Bi_4_O_5_I_2_/Bi_4_O_5_I_2_ product exhibited higher photoactivity with an RhB degradation rate of up to 98.34%, which is higher than the physical mixture material (80.4%), which means that the heterojunction between Bi_4_O_5_I_2_([Hmin]I) and Bi_4_O_5_I_2_(KI) was a chemical force rather than a simple physical connection. Within the Bi_4_O_5_I_2_/Bi_4_O_5_I_2_ nanostructure, the photoinduced transfer of h^+^ and e^−^, the h^+^ accumulated in the VB of Bi_4_O_5_I_2_(KI) (1.45 eV vs. NHE), was insufficient to oxidize H_2_O or OH^−^ to •OH since *E*^0^ (•OH/OH^−^) (2.38 eV vs. NHE). At the same time, the e^−^ accumulated in the CB of Bi_4_O_5_I_2_ ([Hmin]I) (0.54 eV vs. NHE) was more positive than the E^0^(O_2_/•O_2_^−^) (−0.04 eV vs. NHE), which means that no O_2_ was reduced to •O_2_^−^. Therefore, the RhB adsorbed on the nanostructured material is mainly degraded by the h^+^ oxidative degradation accumulated in the VB of Bi_4_O_5_I_2_(KI). In addition, Bi_4_O_5_I_2_ ultra-thin nanosheets synthesized via processing the molecular precursor were found to photo-reduce CO_2_ into CO selectively with a photocatalytic activity of 19.82 μmol h^−1^ g^−1^ [121]. Finally, Yin et al. demonstrated that Bi_4_O_5_I_2_ has good catalytic activity in the degradation of not only MB and RhB but also methyl orange [122].

BiOCl has attracted great interest due to its remarkable photoactivity under UV irradiation [123], and a recent review has focused on this compound [124]. The high photoactivity of BiOCl can be attributed to its unique layered structure, i.e., [Bi_2_O_2_]^2+^ lamellae are interleaved by the double lamellae of Cl atoms, with an internal electrostatic field perpendicular to each layer [125]. This structural feature can effectively promote the transfer of electrons and holes generated inside the crystal face, promote charge separation, and improve quantum yield [126]. However, the broad band gap of BiOCl is 3.1–3.6 eV, which, due to its preparation method and morphology, can only absorb ultraviolet light similarly to TiO_2_, which also limits the effective utilization of solar energy. Therefore, to utilize the high quantum efficiency of BiOCl under visible light irradiation, an efficient approach is combining BiOCl with narrow bandgap semiconductors to form a nanostructure/heterojunction. For example, Li et al. [127] developed BiOCl/Bi_24_O_31_Cl_10_ nanostructures by an ionic liquid self-association method. BiOCl/Bi_24_O_31_Cl_10_ nanostructures were obtained by heating and burning the ionic liquid, which can also be used as the main fuel. The BiOCl/Bi_24_O_31_Cl_10_ nanostructure was applied for the degradation of MO and RhB, which was attributed to the narrow bandgap of 2.3 eV, which enabled efficient electron transfer from the CB of Bi_24_O_31_Cl_10_ to BiOCl and improved the separation efficiency of charge carriers. The BiOCl/Bi_24_O_31_Cl_10_ nanostructure containing 60.4% BiOCl and 39.6% Bi_24_O_31_Cl_10_ exhibited the highest photocatalytic performance. Among these dyes, BiOCl/Bi_24_O_31_Cl_10_ nanostructures exhibited remarkable adsorption activity for cationic dyes of RhB due to their negative surface charges. Furthermore, the main active species responsible for the efficient degradation of pollutants are holes and superoxide radicals involved in the photocatalytic process. Both Bi_4_O_5_Cl_2_ and Bi_4_O_5_l_2_ proved to be efficient for photocatalytic water splitting for hydrogen evolution, but Bi_4_O_5_Cl_2_ is the best for H_2_ production [128].

BiOCl and related compounds are best prepared with an exposed (001) surface to obtain the best activity, owing to the effect of the self-induced electric field by this polar surface [129]. In addition, surface OVs play an important role. The OVs extend the visible-light adsorption range from 200 nm to 800 nm because of the formation of a localized state [130], and they significantly improve the UV-light harvesting ability [131]. In this work, Dong et al. synthesized BiOCl with OVs by solvothermal-induced hot ethylene glycol reduction at 160 °C for 12 h, followed by mixing with H_2_O_2_, drying, and subsequent treatment at 300 °C in an O_2_ atmosphere for 4 h. They demonstrated that the solid solution with OVs increased the removal ratio of toluene from 52.5% to 64% and that of NO from 33.2% to 43.5% in air under 360 nm UV-light irradiation for 15 min. They also showed that the main reason for the important improvement of the toluene degradation comes from the shortening of the toluene degradation pathway via the surface OVs, which enables the production of radicals with high oxidation capability for the accelerated chain scission of the ring-opening intermediates. Zhao et al. showed that the photocatalytic degradation of dyes with Bifocal OVs is greatly enhanced when H_2_O_2_ is added, because the activation of H_2_O_2_ increases the production of •O_2_¯, (H_2_O_2_ + *h*^+^ → •O_2_^−^ + 2H^+^) [130]. Again, the localized state introduced by the OVs is important here, because the electrons in the localized state are transferred to the CB by interband excitation. Then electrons are trapped by O_2_ and generate •O_2_¯. Besides OVs, co-doping is another way to create a localized level in the band gap of BiOCl to expand the light absorption region. Wang et al. prepared co-doped BiOCl nanosheets using a simple hydrothermal route. They exhibited outstanding photocatalytic performance in degrading BPA under visible light irradiation with a degradation rate of 3.5 times higher than that of pristine BiOCl [132].

Jia et al. used diatomite as a solid dispersant to immobilize BiOCl microspheres. BiOCl/60% diatomite presented 94% removal efficiency for CIP under simulated solar light within 10 min irradiation, and also presented a 42.9% total organic carbon (TOC) removal after 240 min, and good reusability [133].

Alansi et al. [134] synthesized the OVs rich in BiOCl_0.8_Br_0.2_ flower-like materials under direct sunlight exposure within 10 min and noted that they changed color from white to black (i.e., UV-BiOCl_0.8_Br_0.2_), which the authors applied for RhB degradation under visible light. When pristine BiOCl_0.8_Br_0.2_ was prepared using low-frequency UV radiation, pristine BiOCl_0.8_Br_0.2_ nanoflowers with a highly exposed facet (001) on the surface were obtained. The (001) facet of pristine BiOCl_0.8_Br_0.2_ has a tight structure, in which the high-density oxygen atoms are exposed with long, weak Bi-O bonds, facilitating the escape of oxygen atoms from the surface, creating OVs behind them. Due to the abundance of OVs of UV-BiOCl_0.8_Br_0.2_, the nanoflowers enhanced photocatalytic activity because the vacancies serve as electron capture centers. The proposed photocatalytic mechanism is as follows: the presence of Br in the BiOCl_0.8_Br_0.2_ material significantly increased its surface area and decreased the Bi-O bond energy on BiOCl, which in turn provides the formation of OVs resulting in wider visible light absorption and a fast charge-transfer rate. Water molecules are rapidly adsorbed on the OV sites of the aqueous solution, and adsorbing on the (001) BiOCl_0.8_Br_0.2_ surface leads to the formation of a layer of hydroxyl groups. Hydroxyl groups increase the length of the Bi-O bond and thus reduce its energy, which provides for the escape of oxygen atoms from the surface upon exposure to an energy source, leaving OVs behind them. Therefore, by exposing pristine BiOCl_0.8_Br_0.2_ to low-energy irradiation, such as the UV component of natural sunlight irradiation, the hydroxyl readily exits and allows the regeneration of OVs on the surface, which changes color from white to black. The weakening of the Bi-O bond in the presence of Br has also been used to synthesize BiOBr ultrathin nanosheets with abundant surface Bi vacancies (VBi-BiOBr) by reactive ionic liquid ([C_16_mim]Br)-assisted synthesis at room temperature [135]. With the advantages of optimized CO_2_ adsorption, activation, and CO desorption, *V*_Bi_-BiOBr UNs can deliver a 3.8-times-improved CO formation rate relative to BiOBr nanosheets, with a selective CO generation rate of 20.1 μmol g^–1^ h^–1^ in pure water. Another example of CO_2_ conversion is given by BiOBr atomic layers with OVs obtained by ultra-sonication exfoliation followed by UV irradiation. The visible-light-driven conversion rate of CO_2_ to CO was increased to 87.4 µmol g^−1^ h^−1^, much higher than the value obtained in the absence of VOs [136].

Wang et al. fabricated OV-rich sulfur-doped BiOBr nanosheets through a facile one-step solvothermal method [137]. The synergistic effect between S doping and oxygen vacancy led to superior photoactivity for non-dye organic contaminants. In particular, under visible light irradiation, the optimal BB-5S sample exhibited 98% degradation efficiency of 4-chlorophenol within 120 min.

The solid solution BiOCl_x_Br_1−x_ exists in the whole range 0 ≤ *x* ≤ 1, so that the ratio of Cl and Br can be tuned to decrease the band gap and thus improve the photocatalytic activity. In particular, Yang et al. prepared this solid solution through a glycol-assisted hydrothermal process [138]. The degradation rate of methyl orange (in aqueous solution) reached a maximum value at *x* = 0.5.

In BiO(ClBr)_(1−x)/2_I_x_ solid solutions, the introduction of I has two advantages. First, the ionic radius of I is larger than that of Cl or Br, so that the introduction of I leads to a lattice dilatation in the c axis, which reduces the energy barrier at the interface of different crystal planes and reduces the recombination rate of the photo-electrons and holes. Second, the CB edge is mainly composed of Bi 6*p* and is thus not significantly dependent on *x*. On the other hand, the valence band edge is mainly composed of the p states of the oxygen and halogens, so it is modified, and actually increases with *x*. Consequently, the band gap decreases with the introduction of I and can be varied from 2.88 to 1.82 eV depending on *x*. As a result, BiO(ClBr)_(1−x)/2_I_x_ showed improved photoactivity for the degradation of 2-propanol to acetone and CO_2_, under visible light [139].

Dong et al. synthesized four-layered bismuth oxyhalides BiO*X* and BiO*X*O_3_ (*X* = Br, I). They found that the order of the photocatalytic performance for water splitting (including the carrier’s lifetime, photocurrent density, and H_2_ evolution rate) is BiOBrO_3_ > BiOI > BiOIO_3_ > BiOBr, emphasizing the role of the polar electric field [140]. The remarkable performance of BiOBrO_3_ is due to the inhibition of the recombination of the charge carriers by the internal polar electric field along the (001) direction. On another hand, the built-in electric field has no impact on the recombination rate in bulk BiO*X*, owing to the mirror symmetry. However, the recombination is hindered in BiOI by the surface polar electric field, which breaks the mirror symmetry. This polar behavior of BiO*X*O_3_ is due to the difference between the crystallographic structure of BiO*X* and BiO*X*O_3_. They all crystallize in the sillenite structure, but the double X^−^ layers in BiO*X* are replaced by double [*X*O_3_] layers in BiO*X*O_3_. The *X*O_3_ units have a trigonal pyramidal structure that generates electric dipoles. For the same reason, BiOIO_3_ is a polar material, with the *c*-axis as the polar axis, which explains its high photocatalytic activity [141,142]. Chen et al. synthesized BiOIO_3_ single crystal nanoribbons along the (001) direction to take advantage of the strong polarity of the IO_3_ units. This polarity acted collaboratively with surface oxygen vacancies to boost CO_2_ reduction [143]. Huang et al. found that the activity of BiOIO_3_ for photocatalytic water splitting can be increased by V^5+^ ion doping into IO_3_ pyramidal units. The •O_2_^−^ and •OH evolution rates of BiOI_0.926_V_0.074_O_3_ increased by ∼3.5- and ∼95.5-fold, respectively, with respect to BiOIO_3_ [144]. Another example of the beneficial effect of the polarization-induced electric field is the modification of porous BiVO_4_ microtubules with inorganic acids. The generation of free hydroxyl radicals by the ionization of hydroxyl groups in the modified inorganic acid increased the intensity of the surface electric field, enhancing their reactivity toward CTC degradation [145].

## 5. Type of Photocatalytic Mechanism of Bismuth-Based Photocatalysts

### 5.1. p–n Junction

The synthesis of nanostructures with highly reactive exposed faces and p–n junctions is of great interest for semiconductor photocatalysis. The construction of nanostructured semiconductor junctions has been very active recently because of their perfect effect in promoting the separation of photogenerated charge carriers and enhancing photocatalytic reactions. In general, nanostructured catalysts containing p–n junctions with direct contact between p-type and n-type semiconductors have drawn much devotion due to their large potential gradients, and the built-in electronic field established at their junction level can induce efficient charge transfer and separation. The main effective strategy to enhance photocatalytic activity is crystal-facet engineering. We have already mentioned the case of BiO*X*/CuFe_2_O_4_ [58]. Another example is provided by Cai et al. [146], who reported nanostructures of β-Bi_2_O_3_/Bi_2_O_2_CO_3_ and α-Bi_2_O_3_ prepared by a rational calcination process of Bi_2_O_2_CO_3_, used as photocatalysts for MB degradation under visible light irradiation. A p–n junction was successfully created by the proposed synthetic procedure. The β-Bi_2_O_3_/Bi_2_O_2_CO_3_ (at 300 °C) nanostructure reduces recombination by promoting the separation of photogenerated electrons and holes, showing higher MB degradation efficiency than Bi_2_O_2_CO_3_ and α-Bi_2_O_3_. When β-Bi_2_O_3_ is in contact with Bi_2_O_2_CO_3_, the CB potential of Bi_2_O_2_CO_3_ is more positive than that of β-Bi_2_O_3_ with the adjustment of the Fermi level. Therefore, electrons generated on β-Bi_2_O_3_ CB can be transferred to Bi_2_O_2_CO_3_ by the electric field formed inside. Therefore, the formation of a p–n heterojunction of β-Bi_2_O_3_/Bi_2_O_2_CO_3_ can effectively separate electron-hole pairs and suppress the undesired recombination of electrons and holes. The separated electron-hole pairs are then freely transferred to the surface to react with the adsorbed dye molecules, thereby enhancing the photocatalytic activity of the nanostructures.

Huang et al. [147] pioneered the in situ-constructed BiOI/Bi_12_O_17_C_l2_ nanostructures consisting of BiOI nanosheets grown vertically on the surface of the Bi_12_O_17_C_l2_ plate, forming a unique front-coupling nanostructure that enables high exposure of the (001) facet reaction exposed surface of BiOI. The photocatalytic behavior of various industrial pollutants such as 2,4-DCP, RhB, BPA, and antibiotics (TCHC) was tested on BiOI/Bi_12_O_17_C_l2_ nanostructures. The BiOI/Bi_12_O_17_C_l2_ nanostructures not only exhibited significantly enhanced photoactivity but also exhibited strong non-selective photooxidation ability under visible light irradiation. The BiOI/Bi_12_O_17_C_l2_ nanostructures exhibit the benefits of facilitating the separation and transfer of charge carriers, which originate from the BiOI (001) active facet and p–n junction responsible for high photoactivity. The highly promoted photoactivity of BiOI/ Bi_12_O_17_C_l2_ nanostructures is mainly credited to the following aspects: (i) Bi_12_O_17_C_l2_ can serve as an excellent substrate to support and uniformly distribute BiOI nanosheets, which helps to increase the specific surface area for enhanced absorption and reaction sites. Nevertheless, the enhanced level of the surface area is lower than the enhanced level of photoactivity. (ii) The main advantage of BiOI/Bi_12_O_17_C_l2_ nanostructures is heterojunction formation, which plays an important role, due to the front-side surface coupling assembly of BiOI/Bi_12_O_17_C_l2_ nanostructures enabling the (001) crystal planes to be more exposed. Due to the strong light absorption ability, a large number of photogenerated carriers will appear under visible light irradiation. (iii) Driven by the strong self-built electric field from the (001) active surface of BiOI, these induced electrons and holes flow from the interior of the BiOI nanosheets, densely migrate to the surface, and then accumulate on opposite surfaces, such as the top and bottom surfaces, respectively (Figure 10).

Therefore, the photoinduced electron-hole pairs of BiOI and BiOI/Bi_12_O_17_C_l2_ also contribute to high photoactivity. p-type BiOI and n-type Bi_12_O_17_C_l2_ can form a stable p–n junction. The CB and VB of BiOI before contact are lower than those of Bi_12_O_17_C_l2_. After the p–n structure is built, the energy level of Bi_12_O_17_C_l2_ decreases, while the energy level of BiOI increases until BiOI and Bi_12_O_17_C_l2_ reach the Fermi level equilibrium. The bottom of the CB of BiOI and the top of the VB can quickly migrate to the bottom of Bi_12_O_17_C_l2_, while the holes generated by the VB of Bi_12_O_17_C_l2_ are transferred to the VB of BiOI. Therefore, the electron-hole pairs are effectively separated at the p–n junction of BiOI/Bi_12_O_17_C_l2_ nanostructures, and the holes accumulated in the Bi_12_O_17_C_l2_ VB directly oxidize the pollutants. The electrons accumulated at the CB of BiOI are further converted into ^•^O_2_^−^ with a strong oxidizing ability, which subsequently induces the decomposition of various pollutants. This work delivered a new avenue for us to design novel nanostructured photoactive materials with integrated p–n junctions and numerous active exposed facets. We have already noted above that BiPO_4_ has a too-large band gap to have good photocatalytic properties in visible light. BiPO_4_ is n-type. A solution is then to modify the surface of BiPO_4_ with BiOBr*_x_*I_1−_*_x_,* which is p-type, to form a p–n heterojunction, and optimize the band gap by the choice of x. The 5% BiPO_4_–BiOBr_0.75_I_0.25_ heterojunction showed the highest photocatalytic activity in the reduction of CO_2_ [148]. After 4 h of visible light irradiation (*λ* > 420 nm), the yield of CO and CH_4_ reached 24.9 and 9.4 μmol g^−1^, respectively.

Tang et al. [149] reported the construction of BiOI/tetrapod-like ZnO whiskers (T-ZnOw). The p–n junction photocatalysts with different Bi/Zn molar ratios were prepared by the in situ precipitation of BiOI on T-ZnOw templates and applied to degrade RhB and oxytetracycline (OTC) under visible-light irradiation. Compared with other samples, the 1:10 nanostructured photocatalysts with different Bi/Zn molar ratios exhibited the highest photoactivity, namely 97.1% RhB and 88% OTC. This is endorsed by the large specific surface area and efficient separation of charge carriers caused by the formation of p–n heterojunctions between T-ZnOw and BiOI. Figure 11a shows the energy bands of BiOI and T-ZnOw before the formation of the BiOI/T-ZnOw nanojunction. BiOI and T-ZnOW have nested energy levels that are not conducive to the transfer of system-generated charge carriers. Since the BiOI/T-ZnOw nanostructure has higher photoactivity and photocurrent response than T-ZnOw and BiOI, it can be inferred that a p–n junction is formed. T-ZnOw is an n-type semiconductor with a Fermi level close to the CB, while BiOI is a p-type semiconductor with a Fermi level close to the VB. When BiOI and T-ZnOw are in close contact, a p–n junction is formed (Figure 11b).

The electrons are transferred from the T-ZnOw near the p–n junction, and at the same time, the holes are transferred from BiOI to T-ZnOw, causing the positive charge region in T-ZnOw to reach equilibrium with BiOI, and the built-in electric field direction from T-ZnOw to BiOI is constructed at the same time. The band positions of BiOI and T-ZnOw shift up and down together with the Fermi level, and photoactivity occurs as follows: (i) BiOI is excited under visible light, causing electrons to move from VB to CB. Then, since the CB of BiOI is more negative compared to that of T-ZnOw, electrons are easily moved to the CB of T-ZNOw from the CB of BiOI. Furthermore, the built-in electric field can provide the migration of photogenerated electrons from BiO to T-ZnOw. The holes remain in the VB of BiOI, which enables the efficient separation of electrons and holes in BiOI. Therefore, efficient charge carrier separation and more electrons and holes can participate in the photocatalytic process, thereby enhancing the photoactivity. The electrons in the CB of T-ZnOw can react with dissolved O_2_ to generate ^•^O_2_^−^, and then generate OH from O_2_^−^ through a reduction process. The holes in the VB of BiOI directly participate in the oxidation of OTC. Therefore, •O_2_^−^, •OH, and h^+^ jointly participate in the degradation of OTC.

Nie et al. [150] developed a p–n junction-derived flower-like CeO_2-δ_ (n-type) coupled to β-Bi_2_O_3_ of p-type (i.e., β-Bi_2_O_3_/CeO_2-δ_) nanostructures via the thermal decomposition of Bi/Ce precursors (Figure 12). This β-Bi_2_O_3_/CeO_2-δ_ nanostructure was used for NO removal under visible light.

The excellent photoactivity of β-Bi_2_O_3_/CeO_2-δ_ nanostructures is attributed to the synergistic effect of oxygen vacancies and p–n junctions. The associated oxygen vacancies not only improve the utilization of visible light and facilitate the separation of electron-hole pairs, but also enhance the adsorption of NO and the activation of O_2_. In fact, the synergistic effect of the p–n heterojunction through the p–n junction favors the interfacial migration of charge carriers, and oxygen vacancies can induce more active radicals. The nanoflower-like β-Bi_2_O_3_/CeO_2-δ_ nanostructures exhibit excellent photoactivity, which can completely remove NO and inhibit NO_2_ production. The authors verified the reaction products by in situ fast Fourier infrared analysis, showing that the main product of nitrate is formed during the photocatalytic process.

### 5.2. n–n Junction

Su et al. [151] developed a post-calcination process for hydrothermal synthesis. One-dimensional, rod-like BiOI/Ag_2_Mo_2_O_7_ nanostructures can reduce the photocatalytic activity of RhB and TC by 70- and 16-fold compared with Ag_2_Mo_2_O_7_, which is attributed to the efficient separation of photogenerated charges. The reason is to form an n–n junction between BiOI and Ag_2_Mo_2_O_7_. From the free radical trapping test results, it can be concluded that •O_2_^−^ and h^+^ species play a major role in photoactivity. In addition, this nanostructure has a photodegradable TC solution, which is basically harmless to Escherichia coli. Figure 13a,b depict the possible photocatalytic mechanism of BiOI/Ag_2_Mo_2_O_7_ nanostructures, showing the energy band positions of BiOI and Ag_2_Mo_2_O_7_ before and after contact. When BiOI is combined with Ag_2_Mo_2_O_7_, an n–n junction is formed at the contact interface. Since the Fermi level of BiOI is lower than that of Ag_2_Mo_2_O_7_, the electrons in BiOI can be transferred to Ag_2_Mo_2_O_7_, thus generating a positive consumption layer on one side of BiOI and a negative consumption layer on the other side of Ag_2_Mo_2_O_7_. When the Fermi levels of the two components reach equilibrium, an internal electric field from BiOI to Ag_2_Mo_2_O_7_ will form at the interface. Under illumination, the photogenerated electrons with strong yield in BiOI CB can reduce O_2_ to O_2_^•−^ (O_2_/O_2_^•−^) (−0.33 eV vs. NHE), and the photogenerated holes in Ag_2_Mo_2_O_7_ VB have strong oxidizing ability, which can direct the oxidation of contaminants to non-toxic products. Therefore, the n–n junction can facilitate the separation of photogenerated charges, thereby enhancing the photocatalytic activity.

### 5.3. Z-Scheme

Li et al. [152] constructed ternary nanostructures composed of AgBr anchored on BiOI/g-C_3_N_4_ nanostructures and applied them to degrade MO (20 mg L^−1^) under visible light irradiation. The MO degradation rate of AgBr/BiOI/g-C_3_N_4_ nanostructures on the nanostructured catalyst reached 93.41% within 120 min, which is attributed to the double Z-type heterojunction between AgBr, BiOI, and g-C_3_N_4_, which has a strong Ag electron capture effect (Figure 14). It was concluded that the main active species was •O_2_^−^, and h^+^ also played a role. A double Z-type electron transfer mechanism is formed between AgBr, BiOI, and g-C_3_N_4_. Under illumination, the electrons accumulated on AgBr can easily react with the attached Ag^+^ to form metallic Ag, so the AgBr/BiOI/g-C_3_N_4_ system is transformed into Ag/AgBr/BiOI/g-C_3_N_4_. Metallic Ag has good electron-trapping ability; it can capture electrons to generate active •O_2_^−^ from the CB of AgBr for degrading MO molecules in solution. However, due to the small doping connection of BiOI and AgBr, the degradation effect is limited. Finally, for the potential of oxidation to •OH (•OH/H_2_O = +1.99 eV and •OH/OH^−^ = +2.4 eV vs. NHE), the g-C_3_N_4_ VB is lower, but it is higher for BiOI and AgBr. Therefore, the amount of •OH produced is small, and it is concluded from the scavenger experiments that •OH can hardly degrade the pollutants during the photocatalytic reaction.

Graphitic carbon nitride (g-C_3_N_4_) has photocatalytic activity for BPA degradation [153,154]. g-C_3_N_4_ is well matched with Bi_2_WO_6_ composites. In particular, the Z-scheme g-C_3_N_4_-Zn/Bi_2_WO_6_ synthesized by a two-step solvothermal method followed by a calcination process, using 2.0 g of dicyanamide as the precursor for g-C_3_N_4_, photodegraded 93% of the BPA within 120 min [155]. Even better results were obtained with Bi_2_WO_6_/g-C_3_N_4_/black phosphorus quantum dots (BPQDs) composites fabricated by the hydrothermal reaction of Bi_2_WO_6_ and g-C_3_N_4_ and a succedent BPQDs modification [156]. This composite with a direct dual Z-Scheme configuration showed photocatalytic activity for BPA degradation in visible light (95.6%, at 20 mg L^−1^ in 120 min), higher than that of Bi_2_WO_6_ (63.7%), g-C_3_N_4_ (25.0%), BPQDs (8.5%), and Bi_2_WO_6_/g-C_3_N_4_ (79.6%), respectively.

Deng et al. [157] fabricated a Z-scheme black BiOCl-Bi-Bi_2_O_3_/rGO heterojunction, where rGO and metallic Bi serve as charge-transfer channels between black BiOCl and Bi_2_O_3_. The black BiOCl-Bi-Bi_2_O_3_/rGO_0.4_ shows the highest visible-light photocatalytic activity with almost complete degradation of 2-nitrophenol, owing to the proper bandgap match between black BiOCl and Bi_2_O_3_, multiple charge-transfer channels via Bi-bridge and rGO, and efficient charge separation.

Recently, Ag/SnO_2-x_/Bi_4_O_5_I_2_ showed high efficiency in degradation and antibacterial properties, owing to the Z-scheme of this ternary composite. The optimum sample of 3% Ag/SnO_2−x_/Bi_4_O_5_I_2_ can degrade 80% TC in 120 min, inactivate Escherichia coli (*E. coil*) in 15 min, and Staphylococcus aureus (*S. aureus*) in 20 min under LED light [158].

Zhang et al. [159] pioneered microwave-hydrothermally synthesized Bi_2_SiO_5_/Bi_2_SiO_20_ nanostructured photocatalysts by bismuth nitrate and nano-SiO_2_ as precursors and applied for the degradation of RhB and MB dyes under UV-light irradiation. The results show that the photocatalytic activities of the Bi_2_SiO_5_/Bi_2_SiO_20_ nanostructures of RhB and MB dyes are 3-fold and 4.3-fold higher than that of Bi_12_SiO_20_, which are attributed to their large specific surface area, smaller particle morphology, and good crystallinity through their heterogeneity. The heterojunction facilitates an efficient charge separation capability. The trapping test results show that superoxide radicals and holes play a major role in the photoactivity. The Bi_2_SiO_5_/Bi_2_SiO_20_ nanostructured photocatalyst has Z-type photocatalytic activity. However, the oxidizing power of the photogenerated holes at the Bi_2_SiO_20_ VB is not sufficient to oxidize H_2_O to •OH because its potential is shallower than that of •OH/H_2_O (2.8 eV vs. NHE). The CBs of Bi_2_SiO_5_ and Bi_2_SiO_20_ are not sufficiently negative compared to the standard reduction potential O_2_/O_2_^•−^ (−0.33 eV vs. NHE), indicating that electrons cannot be captured by O_2_ in solution to form reduced •O_2_^−^. According to the trapping test and ESR results, h^+^ and •O_2_^−^ play important roles in photodegradation; therefore, another common possible mechanism is a Z-type heterojunction. The photogenerated electrons on Bi_2_SiO_5_ transfer from the CB of the photogenerated holes to the VB and can oxidize H_2_O to •OH, while the photogenerated electrons on the CB of Bi_12_SiO_20_ cannot reduce O_2_ to •O_2_^−^. This finding contradicts the test results, indicating that the Z-scheme system also cannot explain the degradation mechanism of Bi_2_SiO_5_/Bi_12_SiO_20_ nanostructures. Surface oxygen vacancies are considered shallow donors for semiconductor photocatalysts and can serve as adsorption and reaction sites, as shown in Figure 15a. For example, oxygen vacancies can dynamically capture directly excited electrons from the CB and can directly activate O_2_ to form reactive oxygen species (^•^O_2_^−^). In fact, Bi_12_SiO_20_ and γ-Bi_2_O_3_ have similar crystal structures, in which 80% of tetrahedral sites are occupied by Bi^3+^ and 20% of vacancies (Si^•^) in the γ-Bi_2_O_3_ crystal structure. Oxygen vacancies in Bi_12_SiO_20_ are mainly concentrated in tetrahedral silicate groups (Si^•^_Bi_O_4_) [160]. Therefore, the photocatalytic mechanism of Bi_2_SiO_5_/Bi_12_SiO_20_ nanostructures can have multiple charge-transfer channels, as shown in Figure 15b. The major contributions are: (i) the existence of surface oxygen vacancies conducive to the enrichment of O_2_; (ii) the photo-induced generation of holes and electrons can control the direction of charge transfer from Bi_12_SiO_20_ to Bi_2_SiO_5_ and also control the electron transfer between O_2_ and Bi_2_SiO_5_/Bi_12_SiO_20_ nanostructures, thereby improving carrier separation in photocatalysis; and (iii) surface oxygen vacancies continuously capture and release electrons to generate new active species, acting as d-electron carriers, which then donate electrons to the anti-bonding orbital of O_2_, reducing it to •O_2_^−^. The relevant reactions involving photocatalytic removal of dyes are as follows:Bi_2_SiO_5_/Bi_12_SiO_20_ + hν → (Bi_2_SiO_5_/Bi_12_SiO_20_ + h^+^) + e^−^(4)
(5)$$O_{2}+e_{\mathrm{OV}}^{-}\to •O_{2^{-}}$$
h^+^(Bi_2_SiO_5_) → h^+^(Bi_12_SiO_20_) (6)
dye + •O_2_^−^ → degradation products(7)
dye + h^+^ → degradation products (8)

However, it remains questionable why •O_2_^−^can still be generated when both Bi_2_SiO_5_ and Bi_12_SiO_20_ have negative CB potentials compared to the O_2_/•O_2_^−^ reduction potential. Finally, the dye is degraded by the interaction of *•*O_2_^−^ and h^+^. This is the unique reaction mechanism of the photocatalytic process proposed in their study.

Other members of the layered bismuth oxide family were also successfully used for the construction of Z-scheme heterojunctions with high photocatalytic activity; in particular, Bi_2_MoO_6_. This is a member of the Aurivillius family, a potential candidate as an excellent photocatalyst and solar-energy-conversion material for water splitting and the degradation of organic compounds under visible-light irradiation [161,162]. We guide the reader to the excellent review on the recent advances in the photocatalytic degradation of organic pollutants using Z-scheme Bi_2_MoO_6_-based heterojunctions [54].

### 5.4. S-Scheme

The S-scheme effect promotes the interface charge transfer and can be used to improve photocatalytic activity. Lu et al. synthesized a Bi_2_O_3_/Bi_2_SiO_5_ p–n heterojunction photocatalysts [163]. The p–n heterojunction was formed by increasing the amount of nano-SiO_2_ precursor, which transformed α-Bi_2_O_3_ into β-Bi_2_O_3_. Dou et al. [164] developed Bi_2_O_3_-related oxygen vacancies coupled with Bi_2_SiO_5_ microspheres to self-assemble to form OVs-Bi_2_O_3_/Bi_2_SiO_5_ heterojunctions via a simple one-pot solvothermal process. The OVs-Bi_2_O_3_/Bi_2_SiO_5_ nanostructures consist of one-micron-sized microspheres for the degradation of MO dyes under visible light. The synergistic effect of Bi_2_O_3_ and Bi_2_SiO_5_ greatly improved the removal rate of MO, and the carrier separation and transfer of the OVs-Bi_2_O_3_/Bi_2_SiO_5_ nanostructure were associated with a ladder mechanism, which endowed the OVs-Bi_2_O_3_/Bi_2_SiO_5_ nanostructure with higher photoactivity as compared to bare Bi_2_O_3_. After the combination of Bi_2_O_3_ and Bi_2_SiO_5_, due to the interfacial electric field gradient in the OVs-Bi_2_O_3_/Bi_2_SiO_5_ nanostructure, the Fermi energies of the two materials are arranged into a new energy band structure at their interface (Figure 16). The photogenerated electrons on the CB of Bi_2_SiO_5_ are transferred to the CB of Bi_2_O_3_. The relatively useless photogenerated electrons on Bi_2_SiO_5_ CB can recombine with relatively useless holes on the Bi_2_O_3_ VB. The holes on the Bi_2_SiO_5_ VB can oxidize H_2_O/OH^−^ to form OH radicals. Therefore, MO can degrade electrons and holes in different spatial regions through •O_2_^−^, •OH, or h^+^ oxidation pathways.

Note that S-scheme heterojunctions have been also made between a Bi-based compound and another compound. An S-scheme heterojunction was formed by depositing Bi_2_O_3_ nanoplates on TiO_2_ nano-fibers [165]. This Bi_2_O_3_/TiO_2_ heterojunction demonstrated good activity to remove phenol under visible light. The S-scheme heterojunction formed by the combination of Bi_2_WO_6_ and a metal–organic framework (NH_2_-MIL-125(Ti)) displayed enhanced photocatalytic activity for the removal of RhB and TC under visible light irradiation [166]. Li et al. [167] fabricated a black phosphorus/BiOBr S-scheme heterojunction by a convenient liquid-phase ultrasound combined with a solvothermal method. The photocatalytic performance of this heterojunction for the TC degradation, oxygen evolution, and H_2_O_2_ production rate of Sol-10BP/BiOBr was 7.8, 7.0, and 2.6 times that of pure BiOBr, respectively. Xie et al. fabricated an S-type g-C_3_N_4_/Bi/BiVO_4_ photocatalyst with the aid of a facile substrate-directed liquid phase deposition route [168]. In addition to the S-scheme effect promoting the interface charge transfer, this structure utilized the SPR effect of bismuth. This effect accelerates the separation of the photo-generated carriers [169], already evidenced in heterojunctions of Bi and BiOCl [21,170,171]. Moreover, the excellent NO removal efficiency observed with nanoparticles of Bi on g-C_3_N_4_ was achieved for the optimized size of the Bi nanoparticles (12 nm) [172]. Owing to these synergetic effects, the g-C_3_N_4_/Bi/BiVO_4_ exhibited superior performance toward artificial carbon cycling.

A recent outstanding example of the efficiency of the S-scheme is the Bi_4_Ti_3_O_12_/ZnIn_2_S_4_ S-scheme heterojunction, which demonstrated an outstanding hydrogen production efficiency of 19.8 mmol h^−1^ g^−1^ under visible light irradiation [173].

## 6. Other Strategies for Enhance the Photocatalytic Activity

### 6.1. Doped-Bismuth-Based Photocatalyst

Chen et al. [174] reported the synthesis of C-N-doped β-Bi_2_O_3_ nanosheets by solvothermal calcination using poly(aniline-co-pyrrole) as C and N sources and applied them to degrade 17α-ethynylestradiol. The photodegradation rate of 17α-ethynylestradiol of C-N doped β-Bi_2_O_3_ nanosheets was 98.86% within 20 min under visible light irradiation, which was attributed to the high specific surface area and hydrophilicity of carbon and C and N doping and the post-induced narrow bandgap in β-Bi_2_O_3_. Shahid et al. [175] reported simple wet-chemically derived gadolinium (Gd)-doped BiFeO_3_ nanoparticles grafted onto rGO using an ultrasonic strategy, and formed Gd-doped BiFeO_3_/rGO nanostructures, which were applied by solar irradiation for the degradation of MB dye. Compared with Gd-doped BiFeO_3_ and bare BiFeO_3_, Gd-doped BiFeO_3_/rGO nanostructures exhibited superior photocatalytic activity. The Gd-doped BiFeO_3_/rGO catalyst removed 87% of MB dyes in 120 min with a rate constant of 0.016 min^−1^, while Gd-doped BiFeO_3_ and bare BiFeO_3_ degraded only 66% (0.008 min^−1^) and 55% (0.003 min^−1^) of the MB dye, due to the synergistic effect of Gd-doping and rGO inclusion, resulting in a red-shift in light absorption. Due to the nanoscale features of the structures, the nanostructures fabricated by this synthesis process can suppress charge carrier recombination and charge-transfer resistance by enhancing electronic conductivity and diffusion properties. Shamin et al. [176] developed a low-temperature hydrothermal synthesis of 10% Gd, Cr-doped Bi_25_FeO_40_ for the degradation of RhB and MB. Gd-Cr-doped Bi_25_FeO_40_ exhibits a low band gap of 1.76 eV and higher photocatalytic degradation performance for RhB and MB, which is attributed to the phase distribution, regular power-like morphology, reduced electron-hole recombination, and lower bandgap. Therefore, Gd-Cr-doped Bi_25_FeO_40_ has been shown to be a compelling energy-saving and low-cost strategy for the preparation of sillenite-phase bismuth ferrite as a promising photocatalyst.

### 6.2. Ligand Modification Strategy

Tien et al. [177] reported that Bi_12_O_17_Cl_2_ nanowires were synthesized by the chlorination of α-Bi_2_O_3_ at 400 °C and consisted of tetragonal structures with a length of 15 μm and a diameter of 400 nm. Bi_12_O_17_Cl_2_ nanowires were prepared by a chlorination method. Typically, α-Bi_2_O_3_ nanowires are directly reacted with HCl and converted into Bi_12_O_17_Cl_2_ nanowires, as follows:6α-Bi_2_O_3_ +2HCl → Bi_12_O_17_Cl_2_ + H_2_O (9)

Bi_12_O_17_Cl_2_ nanowires have red emission at 746 nm and strong green emission at 568 nm at room temperature, which is a hallmark of visible-light-emitting materials for photocatalytic applications, because the synthesized Bi_12_O_17_Cl_2_ nanowires exhibit a narrow bandgap of 2.28 eV.

## 7. Future Prospects and Expectations

Regarding the huge number of research works published in the last years, Bi-based materials have received notable attention as possible active solar photocatalysts and demonstrate high photocatalytic activity when used for the degradation of environmental pollutants. Bismuth-based photocatalysts have proved to be a promising class of materials for a variety of energy- and environment-related applications due to their unique, layered structures, excellent physicochemical properties, and tunable electronic structures. However, a single component of bismuth-based materials often suffers from several inherent disadvantages, including low light-harvesting efficiency, few active sites, and the recombination of charge carriers. To overcome these shortcomings, efforts have been devoted to optimizing the photoactivity of bismuth materials by coupling them with metallic or semiconducting materials to alter their band energies, including rational structural design, compositional tuning, electronic structure tailoring, and interfacial engineering. Furthermore, Bi-based photocatalysts with rationally high catalytic yield still require further alteration to enhance their photocatalytic activity. To satisfy this objective, it was recently revealed that microstructures synthesized in a controlled manner using an appropriate bromide source exhibit improved photocatalytic performance induced by the formation of hierarchical 3D flower-like open petal structures [178]. Controllable morphological features of microparticles have always been an important research topic in material synthesis, which allows us not only to perceive unique features but also to obtain desirable physicochemical properties. For instance, several workers demonstrated the effect of solvent on the morphological characteristics of BiOBr by a solvothermal method and showed that the viscosity of the solvent causes morphological changes in the formation mechanism of photocatalytic materials [179,180,181].

Regardless of the good progress in bismuth-based photocatalysts, the nature of the active sites in these nanostructures remains unclear. It is highly desirable to understand the associated photocatalytic mechanism at the nanoscale level during multiple photocatalytic applications. More attention should be paid to the determination and quantification of active sites by calculation and direct experimental analysis. Further in situ characterization and calculations close to realistic conditions are needed to gain an atomic-level view of the relationship between active sites and photoactivity, which would be a merit for better design and adaptation of bismuth-based photocatalysts. As can be envisaged in this short overview, one of the main approaches to the development of highly efficient photocatalytic material passes through the fabrication of complex heterostructures. Table 1 summarizes recent studies on nanostructured Bi-based photocatalysts for pollutant degradation [56,164,182,183,184,185,186,187,188,189,190,191,192,193,194,195,196,197]. The excellent photocatalytic performance of Bi-based heterojunctions for the degradation of cationic pollutants under visible-light irradiation is superior to that of single sheets, which is ascribed to the efficient charge separation and transfer across the phase junction.

Recently, Bi et al. [184] constructed a Bi_2_WO_6_/ZnIn_2_S_4_ phase junction with a Z-scheme structure showing high photocatalytic activity due to the rapid transfer of carriers, which inhibits the recombination of e^−^ and h^+^. Thus, the phase-junction approach is opening new avenues for the development of efficient photocatalysts for water purification and energy conversion. Highly active S-scheme heterojunctions show outstanding photocatalytic activity, in which both •O_2_^−^ radical attack, H^+^ direct oxidation, and OH oxidation are the processes implicated in the removal of pollutants. Ligand modification based on chlorination is another successful strategy to enhance the photocatalytic activity of BBNs [177].

There is a separate class of bismuth-based photocatalysts—alkaline earth metal (Mg, Ca, Sr, Ba) bismuthates (BiO_3_^–^ or Bi_2_O_6_^2–^)—with bismuth in its pentavalent state [198]. Magnesium bismuthate with the composition MgBi_2_O_6_ is a degenerate semiconductor with a bandgap of only 1.8 eV. Shtarev et al. [199] examined the sillenite structure MgBi_12_O_20_ to probe the effect of the cationic composition of this magnesium bismuthate on its photocatalytic properties.

The photocatalytic activity of Ca_3_Bi_8_O_15_ [200] was estimated from the decomposition of various pollutants, e.g., MO (6.1 × 10^–5^ mol L^−1^), RhB (3.0 × 10^–4^ mol L^−1^), and 4-CP (3.0 × 10^−4^ mol L^−1^) in aqueous media irradiated with visible light (420 nm < λ < 800 nm) at room temperature. The photocatalytic activity of SrBi_4_O_7_ assessed by Yang and coworkers [201] was examined through the decomposition of MG in aqueous media (initial concentrations, 5–50 mg L^−1^) under visible light irradiation, subsequently homogenizing the suspension in the dark to ensure adsorption–desorption equilibrium. Optimal conditions appeared to be 5 mg L^−1^ of MG and 1.5 g L^−1^ of the SrBi_4_O_7_ bismuthate. Under such conditions, the irradiation of the SrBi_4_O_7_/MG aqueous suspension for 3 h caused about 98% degradation and 90% mineralization.

**Table 1 ijms-24-00663-t001:** Recent studies on nanostructured Bi-based photocatalysts for pollutant degradation.

Catalyst	Dosage (g L^−1^)	Pollutant	Dye Concentration (mg L^−1^)	Light Conditions	Degradation Time and Efficiency	Ref.
CNFs/g-C_3_N_4_/BiOBr	3.0	TC	20	300 W, Xe lamp, λ > 400 nm	120 min/86.1%	[182]
Bi_2_MoO_6_/CQDs/Bi_2_S_3_	0.3	TC	20	300 W, Xe lamp, λ > 420 nm	120 min/92.5%	[183]
Bi_2_WO_6_/ZnIn_2_S_4_	0.2	MO	10	300 W, Xe lamp, λ > 420 nm	60 min/97.5%	[184]
OVs-Bi_2_O_3_/Bi_2_SiO_5_	1.0	MO	-	500 W, Xe lamp, λ > 420 nm	7 h/71.8%	[164]
BiVO_4_/Bi_2_S_3_	-	Cr(VI)	-	500 W, Xe lamp, 420 nm filters	-	[185]
BiVO_4_/Bi_2_O_2_CO3	-	RhB	-	-	-	[186]
Bi_25_FeO_40_/Bi_2_Fe_4_O_9_	0.1	RhB	10	-	-	[187]
Bi_2_WO_6_/BiOBr	-	RhB	-	300 W, Xe lamp, λ > 420 nm	-	[188]
Bi_2_WO_6_/ZnFe_2_O_4_	0.05	RhB	50	-	300 min	[189]
BiFeO_3_/TiO_2_	0.5	MB	-	300 W, Xe lamp	120 min/96%	[190]
BiFeO_3_/carbon	0.01	MB	-	300 W, Xe lamp	54 min/54%	[191]
BiFeO_3_-GdFeO_3_	0.01	MB	-	Sunlight	120 min/56%	[192]
BiFeO_3_/Fe_3_O_4_	0.02	MB	-	500 W, halogen lamp	40 min/100%	[56]
MOF-BiFeO_3_	0.02	MB	-	300 W, Xe lamp	120 min/93%	[193]
CuO–BiVO_4_	0.01	MB	-	150 W, Xe lamp	150 min/100%	[194]
Bi_2_MoO_6_–ZnSnO_3_	0.01	MB	-	300 W, Xe lamp	60 min/95%	[195]
BiOBr/Bi_2_O_3_	0.01	MB	-	300 W, Xe lamp	50 min/87%	[196]
BiFeO_3_/Bi_2_WO_6_	0.06	MB	-	500 W, halogen lamp	54 min/75%	[197]

## 8. Concluding Remarks

In summary, this review article further emphasizes the development of and recent advances in bismuth-based photocatalysts and discusses various approaches to improve their photocatalytic performance and associated photocatalytic mechanisms by modifying the band energies and electronic structures of nanostructures or heterojunctions. Particular attention has been devoted to the application of BBNs in the degradation of organic pollutants (i.e., organic dyes and pesticides) and water treatment due to their photocatalytic activity for the formation of C-C and C-S bonds and atom-transfer radical-addition-type reactions. Hazardous dyes in wastewater treated by BBN photocatalysts include methylene blue, rhodamine B, 2,4,6-trichlorophenol, methyl orange, acid orange, acetaminophen, carcinogenic reactive black 5, carbamazepine, malachite green, benzene in aqueous solution, phenol, bisphenol A, and antibiotics. BiO*X* (*X* = Cl, Br, and I) nanoparticles and up-conversion phosphors/BiOBr composites are also efficient catalysts for the degradation of NOx gas. Overall, the fundamental study of the synthesis, characterization, adsorption, and photocatalytic applications of some popular bismuth oxide-based materials have been examined and may be helpful for the development of other metal oxide compounds toward dye degradation under solar illumination and for other environmental and energy-related applications. The introduction of crystal plane tailoring, the creation of porous or hollow structures, and the construction of nanostructured features can enhance the photoactivity of bismuth materials. Various strategies have been applied to further enhance their photoactivity, including double-rich approaches, interfacial engineering, metal doping and SPR effects, and the internal coupling of cocatalysts to host materials for various environmental applications. Although efforts have been made to tune bismuth-based materials and optimize the high performance of their photocatalytic activity, their potential has not been fully exploited. Thus, there are several fundamental insights into the formation of phases and their effects on photocatalytic processes. Current research on the interfacial engineering of bismuth-based materials shows that due to the rich physicochemical structural features of nanostructures that bulk materials do not possess, changing their electronic structures and band energies has a great impact on their photocatalytic efficiency. Therefore, more attention needs to be paid to controlling the preparation of bismuth-based photocatalysts using facile synthetic methods. Although efforts have been made to introduce defects and form heterojunctions to optimize photocatalytic activity, the defect types or doping types present in bismuth materials are almost always oxygen vacancies. These defects can induce additional electronic states and affect the electron transfer rate in nanostructures, implying that an increase in the density of states generally increases the photoactivity of the nanostructures. More efficient methods are needed to construct different types of defects in nanostructures and gain insight into the relationship between defect type and number and photoactivity. Therefore, it is highly desirable to achieve high-efficiency photoactivity through further interfacial engineering of bismuth-based materials.

## Figures and Tables

**Figure 1 ijms-24-00663-f001:**
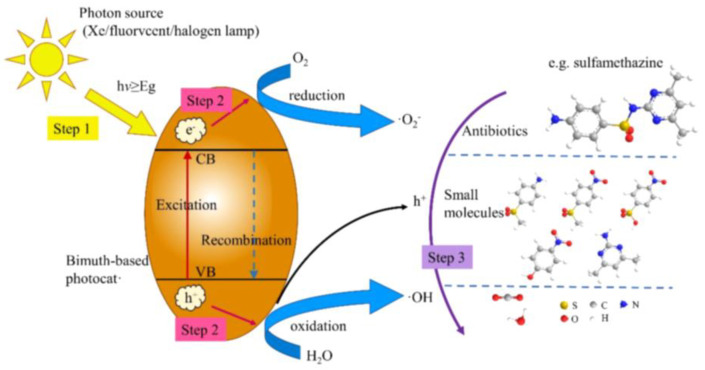
Degradation mechanisms of antibiotics by bismuth-based photocatalysts: absorption of photons with energy higher than the material bandgap, excitation, and reaction. Reproduced with permission from [45]. Copyright 2021 Elsevier.

**Figure 2 ijms-24-00663-f002:**
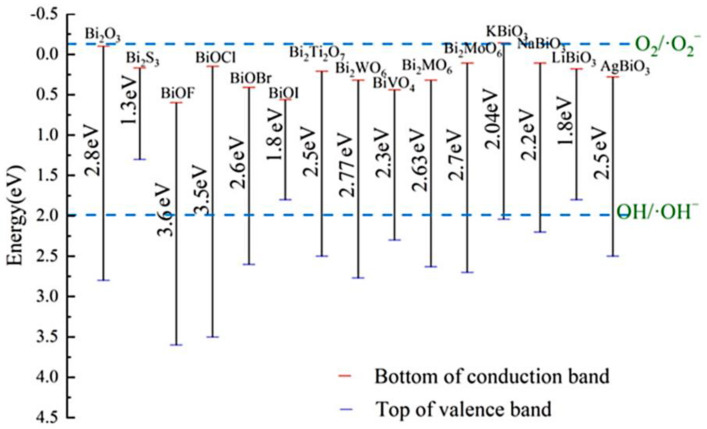
The band edge positions of bismuth-based photocatalysts. Almost all the tops of the valence bands of bismuth-based photocatalysts are higher than the redox potential of •OH/OH^−^ (+1.99 eV). Reproduced with permission from [45]. Copyright 2021 Elsevier.

**Figure 3 ijms-24-00663-f003:**
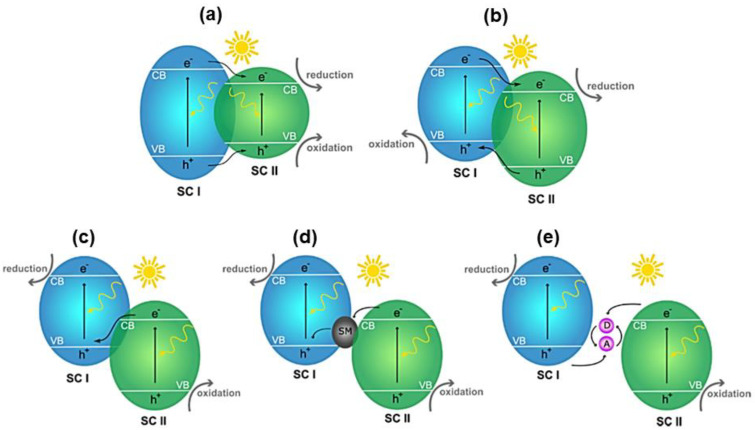
Heterojunction types by band position: (**a**) type I (straddling gap), (**b**) type II (staggered gap), (**c**) direct Z-scheme or mediator-free, (**d**) solid mediator, and (**e**) redox pair mediator. Reproduced with permission from [54]. Copyright 2020 Elsevier.

**Figure 4 ijms-24-00663-f004:**
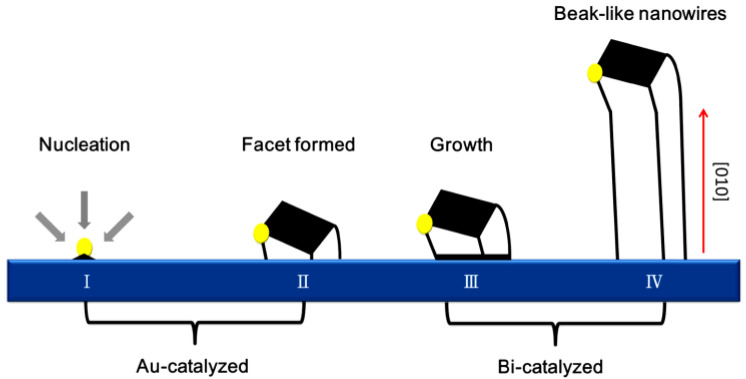
Sketch of growth mechanisms of α-Bi_2_O_3_ nanowires. Processes I–II are driven by the Au-catalyzed mechanism for the formation of the (010) facet; processes III–IV are driven by the Bi-catalyzed mechanism for the formation of beaklike nanowires. Reproduced with permission from [78]. Copyright 2014 Elsevier.

**Figure 5 ijms-24-00663-f005:**
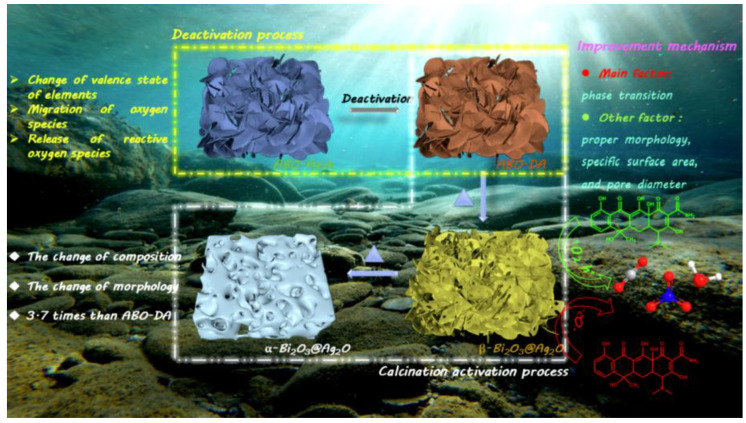
Possible reaction mechanism of the silver–bismuthate system. Reproduced with permission from [99]. Copyright 2022 Elsevier.

**Figure 6 ijms-24-00663-f006:**
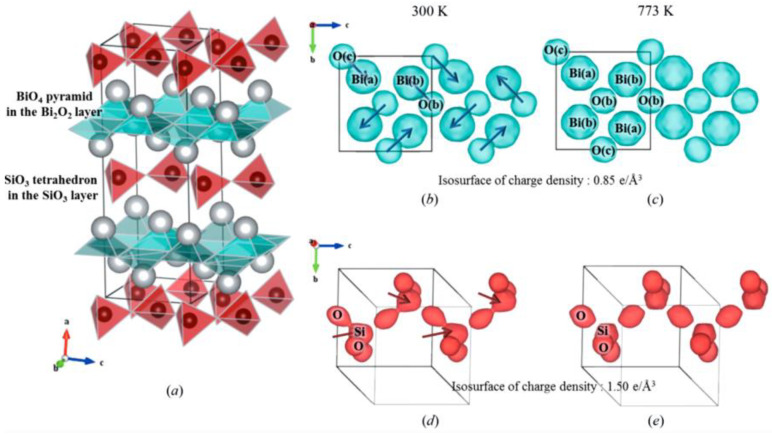
Schematic drawing of the crystal structure of Bi_2_SiO_5_ (**a**) and ECD using the MEM distribution of the Bi_2_O_2_ (**b**,**c**) and the SiO_3_ layer (**d**,**e**) in the ferroelectric (300 K) and paraelectric (773 K) phases. The isosurface of ECD is 0.85 e Å^−3^ and 1.50 e Å^−3^ for the Bi_2_O_2_ and the SiO_3_ layers, respectively. Reproduced with permission from [103]. Copyright 2014 under Creative Commons Attribution (CC-BY) Licence.

**Figure 7 ijms-24-00663-f007:**
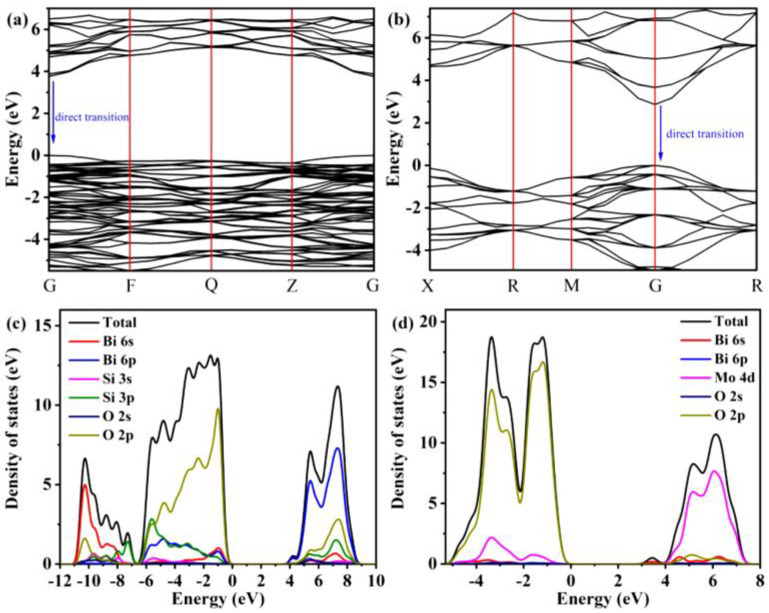
Energy band structures of Bi_2_SiO_5_ (**a**) and Bi_4_MoO_9_ (**b**). Density of states of Bi_2_SiO_5_ (**c**) and Bi_4_MoO_9_ (**d**). Reproduced with permission from [107]. Copyright 2020 under Creative Commons Attribution (CC-BY) Licence.

**Figure 8 ijms-24-00663-f008:**
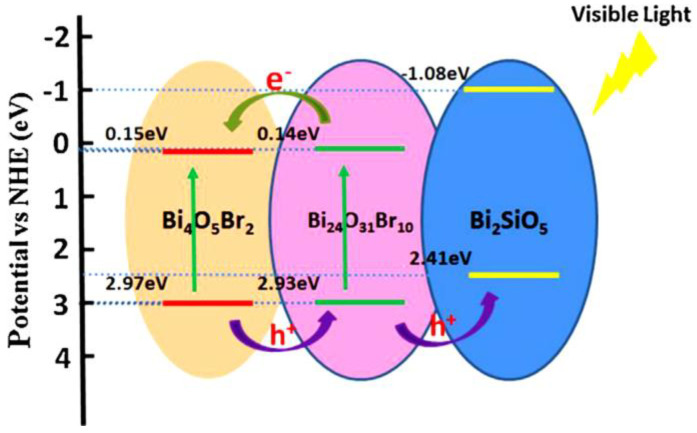
Schematic illustration of the band-gap structure and possible flow of charge carriers through the ternary heterostructure under visible light irradiation. Reproduced with permission from [113]. Copyright 2015 Elsevier.

**Figure 9 ijms-24-00663-f009:**
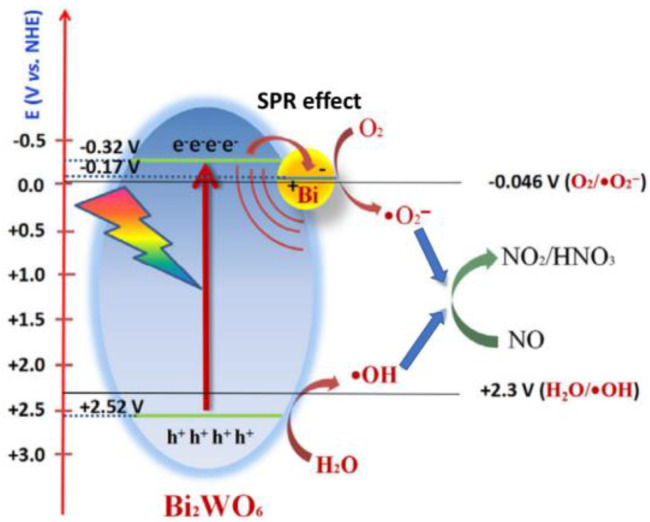
Schematic illustration of the SPR effect on the enhanced photoreactivity of Bi_2_WO_6_ toward NO oxidation after being loaded with Bi nanospheres. Reproduced with permission from [63]. Copyright 2019 Elsevier.

**Figure 10 ijms-24-00663-f010:**
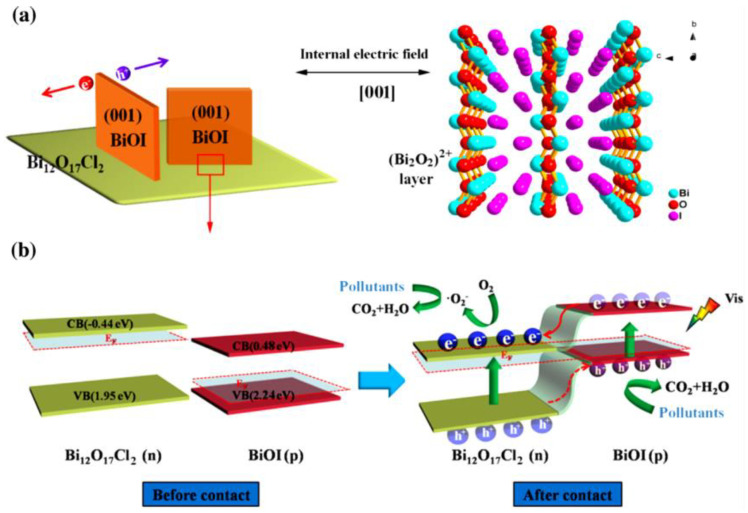
Schematic diagrams for (**a**) the efficient charge separation process through the {001} active facets of BiOI and (**b**) the proposed charge-transfer mechanism via the BiOI@Bi_12_O_17_C_l2_ p–n junction [147]. Copyright 2016 Elsevier.

**Figure 11 ijms-24-00663-f011:**
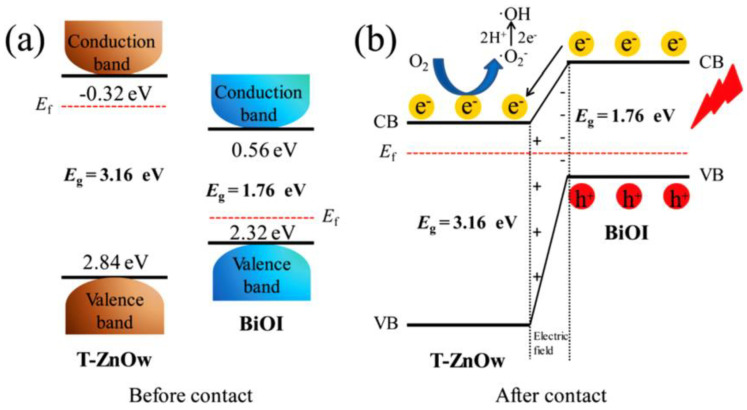
Schematic diagrams of the energy levels of T-ZnOw and BiOI (**a**) before contact and (**b**) after the formation of a p–n BiOI/T-ZnOw heterojunction and possible photocatalytic degradation mechanism under visible light irradiation. Reproduced with permission from [149]. Copyright 2022 The Royal Society of Chemistry.

**Figure 12 ijms-24-00663-f012:**
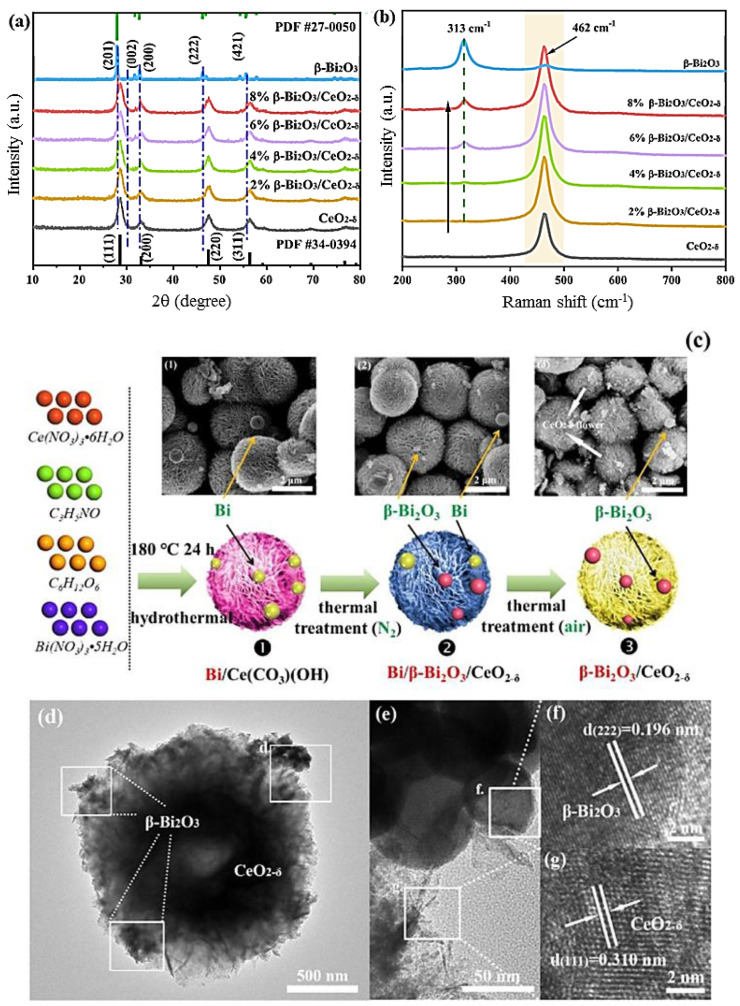
X-ray diffraction patterns (**a**) and Raman spectra (**b**) of CeO_2-δ_, β-Bi_2_O_3_, and β-Bi_2_O_3_/CeO_2-δ_ samples. (**c**) Scanning electron microscopy images and schematic representation of the formation process of 4% β-Bi_2_O_3_/CeO_2-δ_. (**d**,**e**) Transmission electron microscopy (TEM) and high-resolution TEM showing the various species in sample (**f**,**g**) images of 4% β-Bi_2_O_3_/CeO_2-δ_. Reproduced with permission from [150]. Copyright 2021 Elsevier.

**Figure 13 ijms-24-00663-f013:**
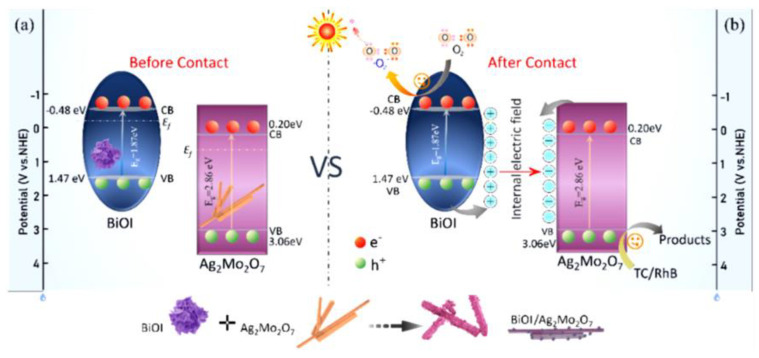
A plausible photocatalytic reaction mechanism diagram for an n–n Ag_2_Mo_2_O_7_/BiOI nanostructure. Reproduced with permission from [151]. Copyright 2022 Elsevier.

**Figure 14 ijms-24-00663-f014:**
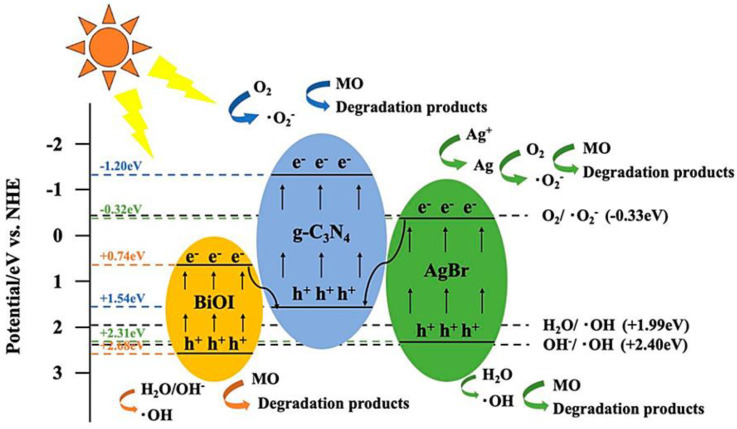
Double Z-type electron transfer mechanism of heterojunction for the photodegradation of MO based on AgBr/BiOI/g-C_3_N_4_. Reproduced with permission from [152]. Copyright 2022 American Chemical Society.

**Figure 15 ijms-24-00663-f015:**
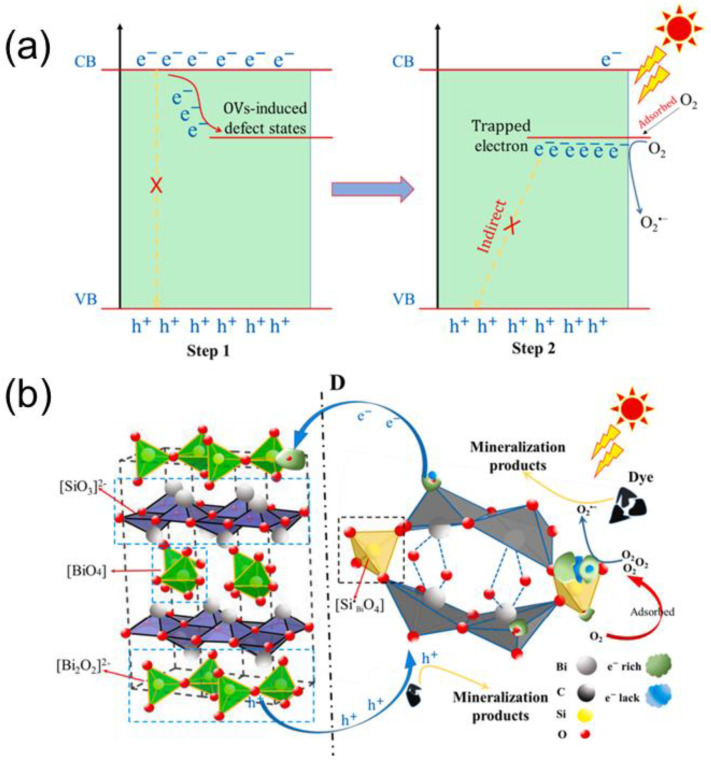
(**a**) Schematic diagram for the enhanced photogenerated electron transfer processes induced by OVs. (**b**) Schematic diagram for the migration and separation of electron-hole pairs and the photocatalytic process of the Bi_2_SiO_5_/Bi_12_SiO_20_ heterojunction photocatalyst: Crystal structure of Bi_2_SiO_5_ (**left**) (Si^•^BiO_4_) tetrahedron and (BiO_5_) pyramids in Bi_12_SiO_20_ crystal structure (**right**). Reproduced with permission from [159]. Copyright 2022 Elsevier.

**Figure 16 ijms-24-00663-f016:**
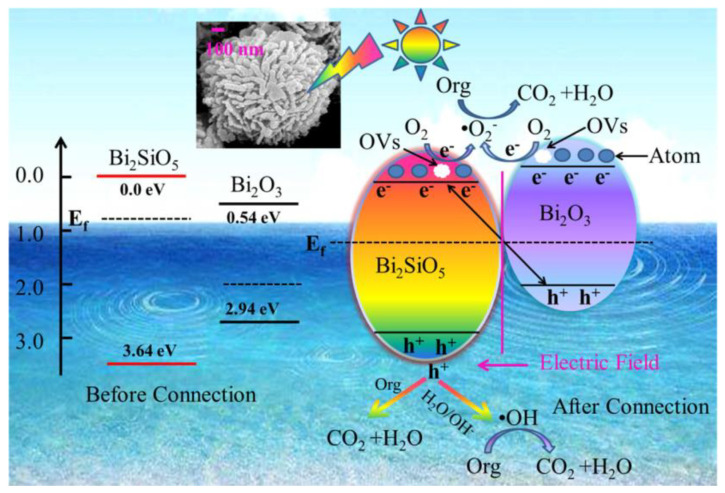
The photo-excited electron-hole separation process over OVs-Bi_2_O_3_/Bi_2_SiO_5_ composite photocatalysts. Reproduced with permission from [164]. Copyright 2020 Elsevier.

## Data Availability

Not applicable.

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
