# Peer review of "High-Throughput Strategies for the Design, Discovery, and Analysis of Bismuth-Based Photocatalysts"

_ijms, 2022, doi:10.3390/ijms24010663_

Round 1

Reviewer 1 Report

This review article further emphasizes the development and recent advances of binary bismuth-based photocatalysts, and discusses various approaches to improve photocatalytic performance and associated photocatalytic mechanisms by modifying band energies and electronic structures of nanostructures or heterojunctions. The author expressed the opinion that it is desirable to achieve high-efficiency photoactivity through further interfacial engineering of bismuth-based materials. I think this paper can be accepted after completing the responses below:

1. Some small errors need to be corrected: such as the blank space between “O2radicals” (line 177) , the tense of the sentence “…showed the degradation is 12 times faster than…” (line 216), etc.

2. The arrangement of the only section “3.1 Synthesis strategies of bismuth ferrites” in “3. Synthesis strategies of bismuth-based photocatalysts” is a little uncoordinated

3. The structure of this review article is weird. The different strategies were emphasized in the Abstract (line19-23), but they were not introduced by this thread in the text.

4. Some backgrounds of the SPR effects in bismuth-based photocatalysts can be introduced in the Background, and the reference can be listed: Chinese Journal of Catalysis 40 (2019) 755–764.

5. Some figures about the photocatalysts with SPR effects can be added in chapter 4.4.

Author Response

We would like to express our sincere gratitude and appreciation to the editor and reviewers for evaluating our manuscript and providing many valuable comments and suggestions, which have helped us greatly to improve the presentation and quality of our manuscript. We have revised the manuscript to address the comments received, as described point-by-point below. All changes are marked in RED color in the revised manuscript.

Please find our reply in the attached sheet.

Reviewer 2 Report

The review article is of great scientific value for researchers involved in the study of bismuth photocatalysts. The review article can be published after the correction of the comments:

1. The use of abbreviations in the text needs to be adjusted.

It is necessary to create a separate list of transcripts for all used abbreviations

abbreviations transcribed several times:

methylene blue  (MB) Line: 209, 343

rhodamine  B  (RhB) Line: 254, 343,

oxygen vacancies (OVs) Line:122,715

bisphenol  A. Line:215,602,845,985

4-tert-butylphenol (PTBP) Line:665,362

ciprofloxacin (CIP) Line: 520,524,738

conduction band (CB) Line 139,513

tetracycline hydrochloride (TC) Line 410 but tetracycline (TC) Line:1004,1094,1097

surface plasmon resonance (SPR) Line:1102, 22, 625

abbreviation - NO on line 357,722, but interpretation on line 924

abbreviation - TEOS on line 545, but interpretation on line 546

not deciphered abbreviations: NHE, DFT, BET, MO, FTIR, XRD, SEM, TEM, HRTEM, UV, LED, ESR

why introduce abbreviations when the abbreviation is not used anywhere else in the text?

acetaminophen (APAP) Line:  289

indigo carmine (IC) Line: 380

ethylenediaminetetraacetic acid (EDTA) Line: 419

brilliant  green  (BG) Line: 430

Advanced oxidation methods (AOPs) Line: 441

electron spin resonance (EPR) Line: 563

(denoted as OVBOC) Line: 718

4-chlorophenol (4-CP) Line: 775

2,4-dichlorophenol (2,4-DCP) Line: 845

Correct the use of abbreviations in the sentence.

Line: 845…such as 2,4-dichlorophenol (2,4-DCP), Rhodamine B, phenol, bisphenol A (BPA), and antibiotics (tetracycline hydrochloride)

2. Uniform designation of bismuth oxide phases

Line 1069 ….which transformed a-Bi2O3  into b-Bi2O3

Line 334, 335 Bandgaps of Bi2O3 polymorphs range in the order of δ-Bi2O3 (3.0 eV) > α-Bi2O3 (2.8 eV) > β-Bi2O3 (2.1 eV) > γ-Bi2O3 (1.64 eV) [84].

3. What is "valence of the valence band"?

Line 514: …while the top of the valence VB of BiPO4

4. What is 1O2 ? Correct if there is a literal error

Line 565 …asOH, h +  and 1O2  played a secondary role

5. What is OH ? Correct if there is a literal error

Line 679  …was insufficient to oxidize H 2O or OH  to OH since E0 (OH/OH) (2.38 eV vs. NHE).

6. What is •O2 ? Correct if there is a literal error

1019 …reduction potential  • O 2 / • O 2-  (-0.33 eV vs. NHE)

956 ….can reduce O 2  to •O 2  (O 2 /•O 2 ) (-0.33 eV vs. NHE)

7. Picture 11 is very poor quality

8. Correct if parentheses are used incorrectly

Line 316  …Ferrite  bismuth  materials  including  perovskites  (BiFeO 3 ,  mullite  Bi 2 Fe 4 O 9 ,  and  sillenite  Bi 25 FeO 40 )  exhibit  outstanding  magnetic,  electronic,  and  dielectric  properties.

9. “2,24-bipyridine” - such a substance does not exist since there are only 12 atoms in bipyridine. Reference 20 lists the substance “2,2'-bipyridine”. Correct if there is a literal error

line 100 …combining 2,24-bipyridine and expensive and not-abundant ruthenium [20] 

10. Specify the range of values “x”

line 355 …and removal of NOx. 

11. With the logical reduction of the term (photocatalyst), it turns out that a quantum of light turns into a photohole and an electron. This may mislead the reader. A photohole is inseparable from a photocatalyst and is a vacancy in its crystal lattice. I propose to change equation (1) to:

photocatalyst + hv → (photocatalyst + h +)  + e

or

photocatalyst + hv → photocatalyst* + h +  + e

line 141 photocatalyst + hv → photocatalyst + h +  + e - ,  

12. Adjust equations (4-7) according to the logic of the photocatalytic process (Line 1043-1047)

How does Equation (4) relate to Equation (1) described earlier?

What is (0-OV  O2-) in equation (5)?

Where does the superoxide radical come from in equation (7)?

13. It is not clear how iron ions with a valence of 2 are formed from a water-insoluble compound (Fe2O3) in which iron is found with a valence of 3?

line 209 …This study shows that the nanointerface promotes the ferrous Fe 2+  ions of Fe 2 O 3

14. Subtitle 2.1. and the text of this subsection describes the mechanism of photodegradation of organic dyes. However, Figure 1 shows the mechanism of antibiotic photodegradation. Correct the text or replace figure 1 in accordance with the text of the subsection.

line 130 …2.1. Fundamental mechanism and main active species for organic dye degradation

line 135 …Figure 1. Degradation mechanisms of antibiotics by bismuth-based photocatalysts

15. The authors use the term "binary photocatalysts" in the subtitle (4) and conclusions (line 1204) to the article. Binary compounds are compounds made up of two different atoms. Photocatalysts from various modifications of bismuth oxide (subheading 4.1.) correspond to this term. However, it is not clear on the basis of what principles the photocatalysts described in the section (for example BiVO4, Bi2SiO5/BiPO4, Bi4O5Br2/Bi24O31Br10/Bi2SiO5) can be called binary. What do the authors mean by bismuth-based binary photocatalysts?

16. There is a separate class of bismuth-based photocatalysts - alkaline earth metal bismuthates. According to photocatalysts, there is a review article (D.S.Shtarev, Nick Serpone A new generation of visible-light-active photocatalysts—The alkaline earth metal bismuthates: Syntheses, compositions, structures, and properties. Journal of Photochemistry and Photobiology C: Photochemistry Reviews Volume 50, March 2022, 100501 https://doi.org/10.1016/j.jphotochemrev.2022.100501). Authors must objectively justify the absence of this class of photocatalysts in their review or supplement their review with information from https://doi.org/10.1016/j.jphotochemrev.2022.100501 

Author Response

We would like to express our sincere gratitude and appreciation to the editor and reviewers for evaluating our manuscript and providing many valuable comments and suggestions, which have helped us greatly to improve the presentation and quality of our manuscript. We have revised the manuscript to address the comments received, as described point-by-point below. All changes are marked in RED color in the revised manuscript.

Please find our reply in the attached file.
